# ELUCIDATING THE DESIGN SPACE OF CLASSIFIER-GUIDED DIFFUSION GENERATION

**Jiajun Ma**[1,2], **Tianyang Hu**[3]*,**Wenjia Wang**[1,2], **Jiacheng Sun**[3]
[1]Hong Kong University of Science and Technology
[2]Hong Kong University of Science and Technology (Guangzhou)
[3]Huawei Noah's Ark Lab

## ABSTRACT

Guidance in conditional diffusion generation is of great importance for sample quality and controllability. However, existing guidance schemes are to be desired. On one hand, mainstream methods such as classifier guidance and classifier-free guidance both require extra training with labeled data, which is time-consuming and unable to adapt to new conditions. On the other hand, training-free methods such as universal guidance, though more flexible, have yet to demonstrate comparable performance. In this work, through a comprehensive investigation into the design space, we show that it is possible to achieve significant performance improvements over existing guidance schemes by leveraging *off-the-shelf* classifiers in a *training-free* fashion, enjoying the best of both worlds. Employing calibration as a general guideline, we propose several pre-conditioning techniques to better exploit pretrained off-the-shelf classifiers for guiding diffusion generation. Extensive experiments on ImageNet validate our proposed method, showing that state-of-the-art (SOTA) diffusion models (DDPM, EDM, DiT) can be further improved (up to 20%) using off-the-shelf classifiers with barely any extra computational cost. With the proliferation of publicly available pretrained classifiers, our proposed approach has great potential and can be readily scaled up to text-to-image generation tasks. The code is available at https://github.com/AlexMaOLS/EluCD/tree/main.

## 1 INTRODUCTION

Diffusion Probabilistic Model (DPM) (Sohl-Dickstein et al., 2015; Ho et al., 2020; Song et al., 2020b) is a powerful generative model that employs a forward diffusion process to gradually add noise to data and generate new data from noise through a reversed process. DPM's exceptional sample quality and scalability have significantly contributed to the success of Artificial Intelligence Generated Content (AIGC) in various domains, including images (Saharia et al., 2022; Ramesh et al., 2022; 2021; Rombach et al., 2022), videos (Ho et al., 2022b; Singer et al., 2022; Ho et al., 2022a; Molad et al., 2023), and 3D objects (Poole et al., 2022; Lin et al., 2023; Wang et al., 2023).

Conditional generation is one of the core tasks of AIGC. With the diffusion formulation, condition injection, especially the classical class condition, becomes more transparent as it can be modeled as an extra term during the reverse process. To align with the diffusion process, Dhariwal & Nichol (2021) proposed classifier guidance (CG) to train a time/noise-dependent classifier and demonstrated significant quality improvement over the unguided baseline. Ho & Salimans (2022) later proposed classifier-free guidance (CFG) to implicitly implement the classifier gradient with the score function difference and achieved superior performance in the classical class-conditional image generation. However, both CG and CFG require extra training with labeled data, which is not only time-consuming but also practically cumbersome, especially when adapting to new conditions. To reduce computational costs, training-free guidance methods have been proposed (Bansal et al., 2023) that take advantage of pretrained discriminative models. Despite the improved flexibility, they have not demonstrated convincing performance compared to CG & CFG in formal quantitative evaluation of guiding diffusion generation. There seems to be an irreconcilable *trade-off* between performance and flexibility and the current guidance schemes are still to be desired.

---

*Correspondence to Tianyang Hu (hutianyang.up@outlook.com)

In this work, we focus on the classical class-conditional diffusion generation setting and investigate the ideal method for guiding the diffusion generation, considering the following criteria: (1) Efficiency with training-free effort; (2) Superior performance in formal evaluation of guided conditional diffusion generation; and (3) Flexibility and adaptability to various new conditions. To this end, we delve into the properties of classifiers and rethink the design space of classifier guidance for diffusion generation. Through a comprehensive investigation both empirically and theoretically, we reveal that: (a) Trained/finetuned time-dependent classifiers have limitations; (b) Off-the-shelf classifiers' potential is far from realized.

While existing methods primarily emphasize classifier accuracy, an ideal classifier should not only provide precise label predictions but also accurate estimations of the gradient of the logarithm of the conditional probability. Given the challenges in efficiently estimating the gradient, classifier *calibration* emerges as a promising alternative, which quantifies how well a classifier recovers the ground truth conditional probability. We show that under certain *smoothness* conditions, a smaller calibration error leads to better estimation of the classifier gradient(Proposition 4.1). Accordingly, we propose the integral calibration error ($\overline{ECE}$) to assess classifier guidance throughout the diffusion reverse process and subsequently, effective pre-conditioning techniques to better prepare the classifier for guidance. Interestingly, our experiments reveal that trained/fine-tuned classifiers (Dhariwal & Nichol, 2021) are less calibrated than off-the-shelf ones when the noise level is high (Figure 1).

Beyond a good probability estimation, an ideal classifier guidance should also integrate seamlessly with the conditional diffusion generation process. Our investigation reveals that the naive implementation of classifier guidance will fade as the diffusion denoising step progresses, resulting in ineffective utilization of the classifier (Figure 4). To address this newly discovered issue, we propose a simple weighing strategy to balance the joint and conditional guidance, which significantly corrects the guidance direction and results in significantly improved sample quality.

To sum up, this work aims to elucidate the design space of classifier-guided diffusion generation. We carry out a comprehensive analysis of the classifier guidance, considering calibration, smoothness, guidance direction, and scheduling. Accordingly, we propose accessible and universal designs that significantly enhance guided sampling performance. Extensive experiments on ImageNet with various DPMs (DDPM, EDM, and DiT) validate our proposed method, showcasing that using off-the-shelf classifiers can consistently outperform both CG and CFG. Additionally, our method can be applied in CFG and further enhance its generation quality. We also demonstrate the scalability and universality of our method in text-to-image scenarios by incorporating CLIP (Radford et al., 2021) guidance with our design. We point out that the operation of increasing recurrent guidance (Bansal et al., 2023) does not fully exploit the potential and comes at the expense of increasing sampling time.

## 2 RELATED WORK

Diffusion models have gained considerable attention due to their potential. Ho et al. (2020); Nichol & Dhariwal (2021); Song et al. (2020a); Peebles & Xie (2022); Karras et al. (2022) demonstrated diffusion models' capacity in generating high-quality samples. Dhariwal & Nichol (2021) introduced fine-tuned time-dependent U-Net (Ronneberger et al., 2015) classifiers to guide diffusion model sampling, resulting in significant improvements. Ho & Salimans (2022) introduced classifier-free diffusion, which has been widely accepted (Rombach et al., 2022; Peebles & Xie, 2022; Ramesh et al., 2022) for generating high-quality samples using both conditional and unconditional diffusion models for inference. In our work, we demonstrate that our off-the-shelf classifier-guided conditional diffusion not only significantly outperforms classifier-free diffusion but also enhances the performance of the classifier-free diffusion model (DiT (Peebles & Xie, 2022)). For guidance in other modalities, Nichol et al. (2021) proposed GLIDE, which utilizes fine-tuned noised CLIP for text-conditioned diffusion sampling. However, it requires the fine-tuning of CLIP on carefully selected noisy data.

In addition, research has explored using off-the-shelf checkpoints to guide diffusion sampling. For example, Wallace et al. (2023) examined the plug-and-play of the classifier guidance, demanding the diffusion architecture to be invertible. Epstein et al. (2023) introduced self-guidance constraints for CLIP-guided sampling in objects' editing. Bansal et al. (2023) applied recurrent guidance operation to universal pretrained models, but only provided demo figures without quantitative evaluation. However, our experiments in Table 1 reveal that increasing the recurrent guidance steps does not significantly

improve the generation quality. In contrast, our calibrated off-the-shelf ResNet[1] significantly enhances the sampling quality (lower Fréchet Inception Distance (FID) (Heusel et al., 2017)) without additional time, highlighting the effectiveness of our proposed guidance scheme. In Graikos et al. (2022), diverse types of regularization constraints are implemented in the diffusion sampling process. They focused on demonstrating the flexibility of diffusion models but the performance limits of each category are not thoroughly tested. Go et al. (2023) propose a multi-expert strategy that involves training multiple classifiers for specific noise value ranges to guide the diffusion sampling process at corresponding time steps. However, this approach requires substantial additional computation, and the experiments in Go et al. (2023) do not provide conclusive evidence that multi-expert-guided sampling can outperform fine-tuned classifier-guided diffusion (Dhariwal & Nichol, 2021). In contrast, our off-the-shelf guided sampling eliminates the need for further training and demonstrates significantly higher sampling performance compared to fine-tuned classifier-guided diffusion (Dhariwal & Nichol, 2021) and classifier-free diffusion (Ho & Salimans, 2022) in formal evaluation.

Table 1: Evaluation of off-the-shelf ResNet with recurrent guidance operation and our calibrated designs in the guided sampling. The ResNet is the official Pytorch ResNet checkpoint; the diffusion model is from Dhariwal & Nichol (2021). We generate 10,000 ImageNet 128x128 samples with 250 DDPM steps for evaluation. Sampling time is recorded as GPU hours on NVIDIA V100.

| Classifier type | recurrent steps | FID | Time (hour) |
|---|---|---|---|
| ResNet | 1 | 7.17 | 14.1 |
| ResNet | 2 | 7.06 | 16.0 |
| ResNet | 3 | 7.14 | 18.0 |
| ResNet (Our-Calibrated) | 1 | 5.19 | 14.1 |

## 3 PRELIMINARIES

**Diffusion model** contains a series of time-dependent model components that apply the forward and reverse processes (Ho et al., 2020; Sohl-Dickstein et al., 2015). Forward process refers to the gradual increment of noise on the original sample $x_0$: $q(x_t|x_0) = \mathcal{N}(x_t; \sqrt{\bar{\alpha}_t}x_0, (1 - \bar{\alpha}_t))$, where $\beta_t$ denotes forward process variance, $\alpha_t = 1 - \beta_t, \bar{\alpha}_t = \Pi_{s=1}^t \alpha_s$. The reverse process refers to gradually generating clean samples from noisy samples: $p_\theta(\hat{x}_{t-1}|\hat{x}_t) = \mathcal{N}(\hat{x}_{t-1}; \mu_\theta(\hat{x}_t, t), \sigma_t)$, where $\mu_\theta(\hat{x}_t, t)$ is derived from removing the diffusion estimated $\epsilon_\theta(\hat{x}_t, t)$ from noisy samples $\hat{x}_t$: $\mu_\theta(\hat{x}_t, t) = \frac{1}{\sqrt{\alpha_t}}(\hat{x}_t - \frac{\beta_t}{\sqrt{1-\bar{\alpha}_t}}\epsilon_\theta(\hat{x}_t, t))$ and $\sigma_t$ denotes the reverse process variance.

**Classifier guidance** (Dhariwal & Nichol, 2021) can be applied in the reverse process for improving generation quality. For conditional diffusion classifier guidance and class $y$, the guided reverse process is adding $\mu_\theta$ with the gradient of the logarithm of the conditional probability: $\mathcal{N}(\hat{x}_{t-1}; \mu_\theta(\hat{x}_t, t) + s\sigma_t \nabla_{\hat{x}_t} \log(p(y|\hat{x}_t)), \sigma_t)$. Specifically, the gradient of the logarithm of classifier $f$ logit in softmax operation $\nabla_{\hat{x}_t} \log(\text{softmax}(f_y(\hat{x}_t)))$ is used for $\nabla_{\hat{x}_t} \log(p(y|\hat{x}_t))$.

**Classifier-free guidance** (Ho & Salimans, 2022) uses the difference between conditional and unconditional noise (score) to represent the conditional probability guidance during the sampling, $\nabla_{\hat{x}_t} \log(p(y|\hat{x}_t)) \propto \epsilon_\theta(\hat{x}_t, y, t) - \epsilon_\theta(\hat{x}_t, \emptyset, t)$. The classifier-free guided sampling becomes $\epsilon_t^* = \epsilon_\theta(\hat{x}_t, y, t) + (s - 1)(\epsilon_\theta(\hat{x}_t, y, t) - \epsilon_\theta(\hat{x}_t, \emptyset, t))$, where $s > 1$ is the classifier-free scale. The unconditional $\epsilon_\theta(\hat{x}_t, \emptyset, t)$ is trained by randomly replacing the class with null class $\emptyset$.

## 4 DESIGN SPACE OF CLASSIFIER GUIDANCE

### 4.1 CLASSIFIERS: FINE-TUNED VS OFF-THE-SHELF

According to Dhariwal & Nichol (2021), a classifier to be used for guiding diffusion generation requires dedicated training or fine-tuning to adapt to noisy images at different time steps during the diffusion process. To implement the time-dependency, the classifier usually employs the downsampling component of U-Net (Ronneberger et al., 2015), and fine-tuning is performed on noisy samples $x_t$ for every $t$ along the forward diffusion process. Such a training procedure is time-consuming (200+ GPU hours for ImageNet 128x128).

---

[1]Pytorch ResNet checkpoints: https://pytorch.org/vision/main/models/resnet.html

In comparison, "off-the-shelf" classifiers refer to the publicly available checkpoints that can be directly deployed without further fine-tuning. The collection of such pretrained classifiers is becoming increasingly more powerful and diverse. There is a line of research, under the umbrella of "model zoo", that specifically studies how to explore and exploit pretrained models for various downstream tasks (Shu et al., 2021; Dong et al., 2022; Chen et al., 2023; Luo et al., 2023). However, when it comes to guiding diffusion generation, "off-the-shelf" classifiers tend to be not robust against Gaussian noise and not adaptable to time-dependent diffusion architectures. Successfully exploiting their knowledge for diffusion models is non-trivial. In our work, we use the official Pytorch ResNet checkpoints as the off-the-shelf classifiers.

As stated earlier, while an ideal classifier for guiding diffusion should provide accurate estimations of $\nabla \log P(y|x)$, calibration error is a promising alternative criterion due to the challenges in efficient gradient estimation. Proposition 4.1 suggests that if a classifier satisfies certain smoothness conditions, a small calibration error indicates a good estimation of the gradient of the log conditional probability.

**Proposition 4.1.** *Let $p \in \mathcal{H}^k(\Omega)$ be the underlying true density function, where $\mathcal{H}^k(\Omega)$ is the Sobolev space with smoothness $k > 1$ defined on a compact and convex region $\Omega$. Assume that there exist constants $c_1, c_2 > 0$ such that $c_2 \geq p(\boldsymbol{x}) \geq c_1, \forall \boldsymbol{x} \in \Omega$. Suppose $p_n$ is an estimate of $p$ such that $\|p_n\|_{\mathcal{H}^k(\Omega)} \leq C$ for some constant $C$ not depending on $n$, where $n$ is the sample size. If $\|p_n - p\|_{L_2(\Omega)} = o_{\mathbb{P}}(1)$, we have $\|\nabla \log p - \nabla \log p_n\|_{L_2(\Omega)} = o_{\mathbb{P}}(1)$.*

To assess the calibration of a classifier and its compatibility with the diffusion model, we propose the integral calibration error $\overline{\text{ECE}}$ (Eq.1) as an estimation of $\int_t \text{ECE}_t$, where $\text{ECE}_t$ (Expected Calibration Error (Naeini et al., 2015)) measures the sample average of the difference between the classifier's accuracy and probability confidence within bins $B$ based on the reverse process sample $\hat{x}_t$ at time $t$.

$$\overline{\text{ECE}} = \frac{1}{k} \sum_{t=0}^{k} \text{ECE}_t, \text{ where } \text{ECE}_t = \sum_{m=1}^{M} \frac{|B_m|}{n} |\text{acc}(B_m(\hat{x}_t)) - \text{conf}(B_m(\hat{x}_t))|. \tag{1}$$

Figure 1 depicts the $\text{ECE}_t$ curves of the two classifiers at different time stages (250 DDPM steps in total). From time 0 to 50, the fine-tuned classifier exhibits lower $\text{ECE}_t$ values compared to ResNet, demonstrating its robustness to Gaussian noise when the images are less noisy. However, as the time steps progress and the noise magnitude increases, the off-the-shelf ResNet achieves lower $\text{ECE}_t$ values than the fine-tuned classifier. This suggests that training on highly noisy or low signal-to-noise samples does not contribute to the fine-tuned classifier guidance. This observation forms the basis that off-the-shelf guidance has the potential to not only match but also surpass the performance of fine-tuned classifiers in our subsequent experiments.

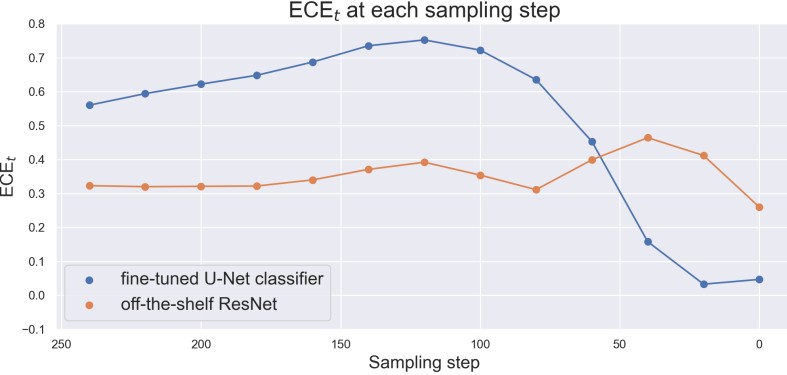

Figure 1: The $\text{ECE}_t$ of the fine-tuned and the off-the-shelf classifiers throughout sampling step.

In the subsequent sections, we explore the design space of classifier-guided diffusion generation to enhance the quality of the classifier gradients along the diffusion process and have better synergy with the diffusion score function: 1. Classifier inputs: to facilitate calibration with the diffusion reverse process, we provide the off-the-shelf classifier with the predicted denoised sample $\hat{x}_0(t)$ during reverse sampling. 2. Smooth guidance: building on Proposition 4.1, we enhance the *smoothness* of the classifier, enabling calibration to result in improved guidance(gradient estimation). 3. Guidance direction: we uncover the classifier guidance diminishes as the diffusion denoising step advances

in Figure 4. To address this, we balance the joint and conditional guidance direction and result in optimal guidance direction. 4. Guidance schedule: we propose a simple yet effective Sine guidance schedule that better aligns with the calibration error curve (Figure 1) of the off-the-shelf classifier.

## 4.2 PREDICTED DENOISED SAMPLES

During the classifier-guided reverse process, there are two types of diffusion-generated sample: the reverse processing sample $\hat{x}_t$ and the predicted denoised sample $\hat{x}_0(t) = (\hat{x}_t - (\sqrt{1 - \alpha_t})\epsilon_\theta(\hat{x}_t, t))/\sqrt{\alpha_t}$ (Song et al., 2020a; Bansal et al., 2023). Considering off-the-shelf classifiers are typically not robust to Gaussian noise and not time-dependent, selecting the appropriate classifier input type is crucial to ensure the best fit. In Table 2, we compare the calibration of two input types: the reverse processing samples $\hat{x}_t$ and the predicted denoised samples $\hat{x}_0(t)$ used in the guided diffusion. Table 2 demonstrates that the off-the-shelf ResNet classifier achieves better calibration when provided with the denoised sample $\hat{x}_0(t)$ compared to $\hat{x}_t$. This improvement in calibration enhances the guided sampling quality with a smaller FID.

Table 2: Comparison of classifier inputs in guided sampling with respect to $\overline{\text{ECE}}$ and FID. Denote the guidance of classifier $f$ as $\text{Guidance}(x) := \nabla_x \log(\text{softmax}(f_y(x)))$.

|  | GUIDANCE$(\hat{x}_t)$ | GUIDANCE$(\hat{x}_0(t))$ |
|---|---|---|
| $\overline{\text{ECE}}$ | 0.36 | 0.28 |
| FID | 8.61 | 7.17 |

## 4.3 SMOOTH CLASSIFIER

A smooth classifier is important to guided diffusion generation. On one hand, Proposition 4.1 indicates that the smoothness of the classifier is key to ensuring good gradient estimation. On the other hand, gradient-based optimization also benefits from increased smoothness. For better guidance, we propose to enhance the smoothness of the classifier. According to Zhu et al. (2021), the Softplus activation, $\text{Softplus}_\beta(x) = \frac{1}{\beta} \log(1 + \exp(\beta x))$, is effective in smoothing the classifier gradient. As parameter $\beta$ approaches infinity, the Softplus function converges to the ReLU activation function. Table 3 and Figure 2 demonstrate that as $\beta$ decreases, the $\overline{\text{ECE}}$ decreases as well, indicating that smoother activation benefits classifier calibration. Consequently, the well-calibrated design enhances the guided sampling performance compared to the baseline (ReLU), reducing FID from 7.17 to 6.61.

Table 3: Ablation study of $\beta$ in Softplus with respect to integral calibration error $\overline{\text{ECE}}$ and FID.

|  | RELU | SOFTPLUS($\beta$=8) | SOFTPLUS($\beta$=5) | SOFTPLUS($\beta$=4) | SOFTPLUS($\beta$=3) |
|---|---|---|---|---|---|
| $\overline{\text{ECE}}$ | 0.34 | 0.31 | 0.26 | 0.21 | 0.07 |
| FID | 7.17 | 6.99 | 6.89 | 6.73 | 6.61 |

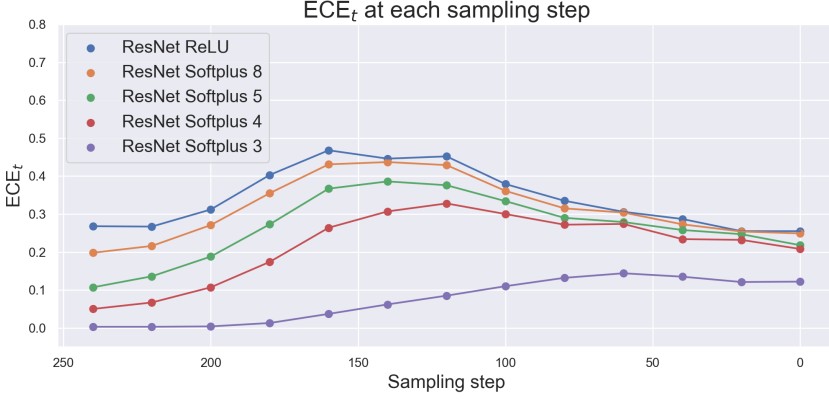

Figure 2: The $\text{ECE}_t$ curves of the off-the-shelf ResNet with ReLU and Softplus activation functions.

### 4.4 JOINT VS CONDITIONAL DIRECTION

In this section, we demonstrate that properly weighing the joint and conditional guidance can significantly improve sampling quality. In Dhariwal & Nichol (2021), the classifier guidance is defined as the gradient of the conditional probability, which can be interpreted as the gradient of the joint and marginal energy (Grathwohl et al., 2019) difference, shown in Eq.2 and 3 as

$$\log p_\tau(y|x) = \log \frac{\exp(\tau f_y(x))}{\sum_{i=1}^N \exp(\tau f_i(x))} = \tau f_y(x) - \log \sum_{i=1}^N \exp(\tau f_i(x)) := -E_\tau(x,y) + E_\tau(x),$$

$$(2)$$

$$\nabla_x \log p_\tau(y|x) = -\nabla_x E_\tau(x,y) + \nabla_x E_\tau(x) \tag{3}$$

To gain a deeper understanding of the guidance direction, we conduct a closed-form analysis in mixed-Gaussian scenarios. Proposition 4.2 provides the derived closed-form joint and conditional directions in mixed-Gaussian scenarios: for conditional probability gradient, the guidance direction is a combination of class mode differences; while the joint probability gradient directly targets the objective class mode $\mu_l$. Figure 3 provides a visual illustration of the two types of guidance directions.

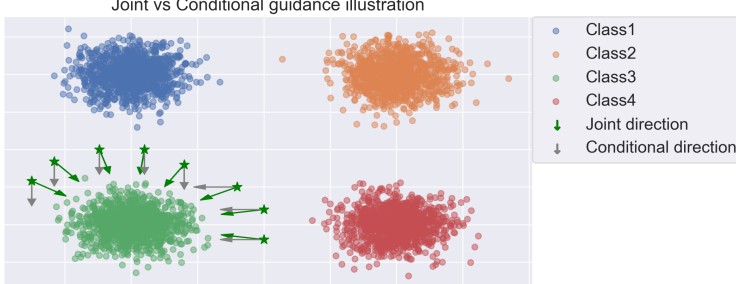

Figure 3: Mixed-Gaussian settings of joint and conditional probability guidance toward Class3.

**Proposition 4.2.** *Let $X \sim P$ be a random variable defined on $\mathbb{R}^d$, with the density function $f(\boldsymbol{x}) = \sum_{k=1}^K b_k f_k(\boldsymbol{x})$, where $f_k(\boldsymbol{x})$ is a normal density function with mean $\boldsymbol{\mu}_k$ and covariance matrix $\boldsymbol{\Sigma}_k$, and $b_k > 0$ with $\sum_{k=1}^K b_k = 1$. Let $Z \in \{1, ..., K\}$ be a random variable satisfying $P(Z = l, X = \boldsymbol{x}) = b_l f_l(\boldsymbol{x})$. Then we have*

$$\nabla_{\boldsymbol{x}} P(Z = l | X = \boldsymbol{x}) \propto \sum_{k=1}^K b_k e^{-\frac{1}{2}(\boldsymbol{x}-\boldsymbol{\mu}_k)^\top \boldsymbol{\Sigma}_k^{-1}(\boldsymbol{x}-\boldsymbol{\mu}_k)} (\boldsymbol{\Sigma}_l^{-1}(\boldsymbol{x} - \boldsymbol{\mu}_l) - \boldsymbol{\Sigma}_k^{-1}(\boldsymbol{x} - \boldsymbol{\mu}_k)),$$

$$\nabla_{\boldsymbol{x}} P(Z = l, X = \boldsymbol{x}) \propto \boldsymbol{\Sigma}_l^{-1}(\boldsymbol{\mu}_l - \boldsymbol{x}).$$

Proposition 4.2 reveals that if all $\boldsymbol{\Sigma}_k$ are identity matrices, the gradient of the joint distribution is simply $\boldsymbol{\mu}_l - \boldsymbol{x}$, directing towards the mode of density $f_l(\boldsymbol{x})$; while the gradient of the conditional distribution is $\sum_{k=1}^K b_k e^{-\frac{1}{2}(\boldsymbol{x}-\boldsymbol{\mu}_k)^\top(\boldsymbol{x}-\boldsymbol{\mu}_k)}(\boldsymbol{\mu}_k - \boldsymbol{\mu}_l)$, which may point to low-density region.

To strengthen the joint $f_y(x)$ guidance (joint energy $E_{\tau_1}(x,y)$), we reduce the value of marginal temperature $\tau_2$ relative to the joint temperature $\tau_1$, shown in Eq.4. The ablation study in Table 4 validates the improvement in the sampling quality by weighing the joint & marginal guidance.

$$\nabla_x \log p_{\tau_1,\tau_2}(y|x) = \nabla_x(\tau_1 f_y(x) - \log(\sum_{i=1}^N \exp(\tau_2 f_i(x)))) := -\nabla_x E_{\tau_1}(x,y) + \nabla_x E_{\tau_2}(x).$$

$$(4)$$

In addition to quantitative metrics, we visually demonstrate the impact of enhancing joint guidance on classifier-guided sampling. Figure 4 displays the intermediate sampling images and the classifier gradient figures over 250 DDPM steps. Figure 4 (a) represents the traditional conditional probability settings ($\tau_1 = 1, \tau_2 = 1$): the classifier gradient figure gradually fades from t=50 to 0, indicating a loss of dog depiction guidance during the sampling. In contrast, Figure 4 (b) showcases strengthened joint guidance ($\tau_1 = 1, \tau_2 = 0.5$): the classifier gradient figure increasingly highlights the dog outline,

Table 4: Ablation study of marginal logit temperature $\tau_2$ with respect to FID. $\tau_1$ is fixed as 1.

| $\tau_2$ | 1.0 | 0.8 | 0.7 | 0.5 | 0.3 |
|---|---|---|---|---|---|
| FID | 6.20 | 5.62 | 5.45 | 5.27 | 5.30 |

providing consistent and accurate guidance direction throughout the entire sampling process. It aligns with Proposition 4.2, highlighting that the gradient of the conditional distribution may point to a low-density region, while the jointly amplified gradient targets the mode of density more precisely.

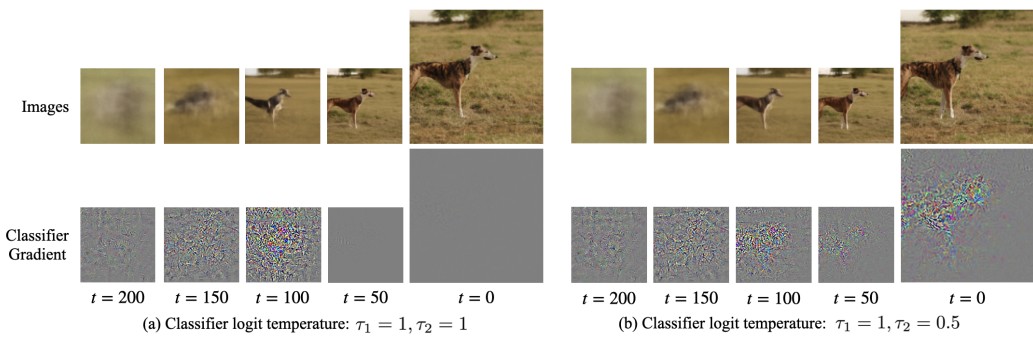

(a) Classifier logit temperature: $\tau_1 = 1, \tau_2 = 1$   (b) Classifier logit temperature: $\tau_1 = 1, \tau_2 = 0.5$

Figure 4: The visualization of intermediate sampling images and classifier gradient figures under conditional probability and joint-strengthened guidance over 250 DDPM steps.

### 4.5 GUIDANCE SCHEDULE

In Dhariwal & Nichol (2021), the classifier guidance scale schedule employs a linear timely-decay variance $\sigma_t = \beta_t$ (Ho et al., 2020). To fully leverage the benefits of a well-calibrated classifier, we introduce a sin factor $\gamma$ to the guidance schedule: $\gamma_t = \sigma_t + \gamma\sigma_T \cdot \sin(\pi t/T)$, where $\sigma_t$ denotes the variance at time $t$. This design choice is motivated by the observation in Figure 1, where the off-the-shelf classifier exhibits consistently lower and more stable $ECE_t$ values during the large noise period (from the beginning to the middle of the reverse process). Consequently, we can amplify the guidance scale during this period to better exploit its effectiveness. The parameter $\gamma$ is used for controlling the magnitude of the guidance amplifying: the bigger the parameter $\gamma$, the greater the time-dependent sin value added to the guidance schedule. Figure D.1 in Appendix A demonstrates the original schedule and the $\gamma$ added schedule. The impact of the sin factor $\gamma$ is examined in the ablation study presented in Table A.3 of Appendix A.

### 5 EXPERIMENTS

### 5.1 OFF-THE-SHELF GUIDANCE FOR DDPM

Guided diffusion (Dhariwal & Nichol, 2021) demonstrates that incorporating a fine-tuned U-Net classifier can significantly enhance image generation quality. However, the classifier is demanded to be a time-dependent U-Net, and the fine-tuning process is time-consuming (200+ GPU hours for ImageNet128x128 classifier fine-tuning). In our approach, we utilize off-the-shelf PyTorch ResNet-50 and ResNet-101 checkpoints with our calibrated design to directly guide the diffusion sampling. Table 5 confirms that our calibrated off-the-shelf ResNet-50 (FID: 2.36) and ResNet-101 (FID: 2.19) not only improve the diffusion baseline quality (FID: 5.91) but also outperforms the fine-tuned classifier guided diffusion (FID: 2.97) and the classifier-free diffusion (Ho & Salimans, 2022) (FID: 2.43) by a significant margin. By leveraging off-the-shelf classifiers, we integrate external knowledge into conditional diffusion models, surpassing existing approaches. The guided sampling algorithm is outlined in Algorithm 1 and the hyper-parameter settings can be found in Appendix C.3.

Table 5: The comparison of the baseline DDPM diffusion, the fine-tuned classifier-guided, classifier-free diffusion, and the off-the-shelf ResNet guided sampling. All models generate 50,000 ImageNet 128x128 samples with 250 DDPM steps.

| IMAGENET 128x128 | CLASSIFIER | IS | FID |
|---|---|---|---|
| DIFFUSION BASELINE (DHARIWAL & NICHOL, 2021) | - | - | 5.91 |
| DIFFUSION FINETUNE GUIDED (DHARIWAL & NICHOL, 2021) | FINE-TUNE | 182.69 | 2.97 |
| CLASSIFIER-FREE DIFFUSION (HO & SALIMANS, 2022) | - | 158.47 | 2.43 |
| DIFFUSION RESNET50 GUIDED (OURS) | OFF-THE-SHELF | 183.51 | **2.36** |
| DIFFUSION RESNET101 GUIDED (OURS) | OFF-THE-SHELF | 187.83 | **2.19** |

---

**Algorithm 1** DDPM off-the-shelf classifier guided sampling.

---

**Parameter:** SoftPlus activation $\beta$, joint logit temperature $\tau_1$, marginal logit temperature $\tau_2$. classifier guidance scale $\gamma_t$.

**Required:** Diffusion model $D_\theta$, variance schedule $\sigma_t$, class label $y$, reverse process sample $\hat{x}_t$, predicted denoised sample $\hat{x}_0(t)$, reverse process noise $\epsilon_\theta(\hat{x}_t, y, t)$, classifier logit of input $x$ on class $y$: $f_y(x)$.

$\hat{x}_T$ sampled from $\mathcal{N}(0, \mathbb{I})$

**for** $t \in \{T, ..., 1\}$ **do**

   $\mu, \epsilon_\theta(\hat{x}_t, y, t) \leftarrow D_\theta(\hat{x}_t, y, t)$

   $\hat{x}_0(t) \leftarrow (\hat{x}_t - (\sqrt{1 - \alpha_t}\epsilon_\theta(\hat{x}_t, y, t)))/\sqrt{\alpha_t}$  ▷ get predicted denoised sample

   $g \leftarrow \nabla_{\hat{x}_0(t)} \log(\exp(\tau_1 f_y(\hat{x}_0(t)))/(\sum_{i=1}^{N} \exp(\tau_2 f_i(\hat{x}_0(t)))))$  ▷ classifier gradient guidance

   $\hat{x}_{t-1} \leftarrow$ sample from $\mathcal{N}(\mu + \gamma_t g, \sigma_t)$

**end for**

**return** $\hat{x}_0$

---

## 5.2 OFF-THE-SHELF GUIDANCE FOR EDM

In this section, we demonstrate the effectiveness of off-the-shelf classifier guidance in fewer sampling steps based on EDM (Karras et al., 2022) model. EDM utilizes a sampling trajectory based on the sampling curvature $d_t = dx/dt$, enabling efficient and high-quality image generation. Our EDM guided-sampling algorithm is outlined in Algorithm 2 of Appendix C.2, where the normalized sample $\hat{x}_i/\|\hat{x}_i\|_2$ is used as the classifier guidance inputs $g = \nabla \log(\text{softmax} f(\hat{x}_i/\|\hat{x}_i\|_2))$, then the gradient $g$ is normalized to align with the sample $\hat{x}_i$ and curvature $d_i$: $\hat{x}_{i-1} \leftarrow \hat{x}_i + (t_i - t_{i-1})d_i + \gamma_i(g/\|g\|_2)$. In our experiments, we present the results of off-the-shelf classifier guidance of ODE sampling on ImageNet 64x64 in Table 6, with the sampling time recorded as GPU hours. The results of 256 steps of SDE sampling (Kingma & Gao, 2023) can be found in Table C.1 of Appendix C.4.

Table 6: EDM baseline and the off-the-shelf ResNet guided EDM sampling. Sampled for mutiple ODE steps. We generate 50,000 ImageNet 64x64 samples for evaluation.

| IMAGENET 64x64 | CLASSIFIER | FID | STEPS | TIME(HOUR) |
|---|---|---|---|---|
| EDM BASELINE | - | 2.35 | 36 | 8.0 |
| EDM RES101 GUIDED | OFF-THE-SHELF | **2.22** | 36 | 8.4 |
| EDM BASELINE | - | 2.54 | 18 | 4.0 |
| EDM RES101 GUIDED | OFF-THE-SHELF | **2.35** | 18 | 4.1 |
| EDM BASELINE | - | 3.64 | 10 | 2.1 |
| EDM RES101 GUIDED | OFF-THE-SHELF | **3.38** | 10 | 2.2 |

## 5.3 OFF-THE-SHELF GUIDANCE FOR DiT

In this part, we showcase the applicability of off-the-shelf guidance in enhancing latent-spaced classifier-free diffusion models, specifically DiT (Peebles & Xie, 2022). DiT stands out in several aspects. Firstly, it utilizes transformer architecture instead of U-Net. Secondly, DiT operates in the latent space, which is encoded by a variational autoencoder (VAE) (Kingma & Welling, 2013) from Stable Diffusion (Rombach et al., 2022). Lastly, DiT is trained in classifier-free setup (Ho & Salimans,

2022). Our DiT guided-sampling algorithm is outlined in Algorithm 3 of Appendix C.2, and the guided performance is presented in Table 7. Unlike Wallace et al. (2023), our off-the-shelf classifier guidance does not require retraining a latent-space-based classifier and imposes no requirements on the diffusion architectures. Notably, there are two key designs in the Algorithm 3:

1. To integrate the pixel-spaced classifier $f$ into latent sampling, we consider the guidance $g$ as the gradient of composite functions, specifically the VAE decoder $V_D$ within the classifier $f$. It can be expressed as: $g = \nabla_{\hat{z}_0(t)} \log(\text{softmax} f(V_D(\hat{z}_0(t))))$.

2. To incorporate classifier guidance $g$ into classifier-free sampling, we normalize the guidance and add the normalized $\bar{g}$ to the conditional and unconditional noise difference. The formula is as follows: $\epsilon_\theta(\hat{z}_t, c, t) + (s - 1)(\epsilon_\theta(\hat{z}_t, c, t) - \epsilon_\theta(\hat{z}_t, \emptyset, t) + \gamma_t \bar{g})$.

Table 7: DiT baseline and the off-the-shelf ResNet guided DiT sampling, sampled for 250 DDPM steps. We generate 50,000 ImageNet 256x256 samples for evaluation.

| IMAGENET 256x256 | CLASSIFIER | FID | PRECISION | RECALL |
|---|---|---|---|---|
| DiT BASELINE | - | 2.27 | 0.828 | 0.57 |
| DiT RESNET101 GUIDED | OFF-THE-SHELF | **2.12** | 0.817 | 0.59 |

## 5.4 CLIP GUIDED DIFFUSION

In this section, we utilize off-the-shelf CLIP (Radford et al., 2021) to guide the conditional diffusion model (Dhariwal & Nichol, 2021) in generating images based on a given prompt (see Eq. D.1 in Appendix D). Compared to approaches in Bansal et al. (2023); Wallace et al. (2023), our text-to-image sampling method achieves more efficient and high-quality samples. Our method does not require recurrent guidance iteration within a single step, resulting in a sampling speed that is approximately 5 times faster than the methods in Bansal et al. (2023). In terms of generation quality, our proposed design leads to significant improvements in CLIP scores in Figure D.6 of Appendix D. Refer to Figure 5 and Figure D in Appendix D for more demonstration figures.

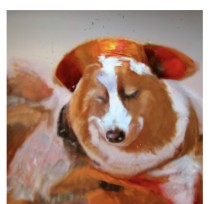 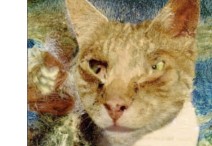 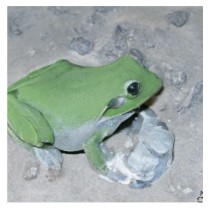 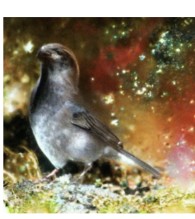

"Party hat on corgi in oil painting"    "Van Gogh Style Cat"    "Ice Frog"    "Bird in the universe"

Figure 5: CLIP guided figures

## 6 DISCUSSION

In this work, we elucidate the design space of off-the-shelf classifier guidance in diffusion generation. Using our training-free and accessible designs, off-the-shelf classifiers can effectively guide conditional diffusion, achieving state-of-the-art performance in ImageNet 128x128. Our approach is applicable to various diffusion models such as DDPM, EDM, and DiT, as well as text-to-image scenarios. We believe our work contributes significantly to the investigation of the ideal guidance method for diffusion models that may greatly benefit the booming AIGC industry.

There are multiple directions to extend this work. First, we primarily investigated classifier guidance in diffusion generation while there are more sophisticated discriminative models, e.g., detection models and visual Question Answering models in other domains. Second, we only considered image generative models, and extending to language models or other modalities would also be a promising direction. We believe that our proposed designs and calibration methodology hold potential for diverse modalities and we leave this for future work.

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
