APPENDIX

# A   CLASSIFIER DESIGN SPACE

## A.1   INTEGRAL CALIBRATION

To justify how the ECE calibration can help improve the guided sampling, we conduct a comparative analysis in different settings of classifiers in guided sampling in Table A.1 based on Figure 1. In addition to the fine-tuned classifier and the off-the-shelf ResNet options, we introduce the ResNet&fine-tuned classifier combination. This combination utilizes ResNet guidance from 250 to 50 timesteps and fine-tuned classifier guidance from 50 to 0 timesteps, resulting in a lower ECE curve over time. The ResNet&fine-tuned combination demonstrates that lower ECE calibration error leads to improved guided sampling quality (lower FID(Heusel et al., 2017)). (Note: the ResNet&fine-tuned combination is used for demonstration purposes only, and we will solely employ the off-the-shelf ResNet in the subsequent analysis and guided sampling). We use the official ResNet checkpoints as the off-the-shelf classifier.

Table A.1: Comparative analysis of different classifier choices in guided sampling. The fine-tuned classifier is from Dhariwal & Nichol (2021) and the ResNet is the official Pytorch ResNet checkpoint ; the diffusion model is from Dhariwal & Nichol (2021) and the dataset is ImageNet 128x128. Generating 10000 samples with 250 DDPM steps for evaluation.

| CLASSIFIER TYPE | FID |
|---|---|
| FINE-TUNED | 8.27 |
| RESNET | 7.17 |
| RESNET&FINE-TUNED | 6.94 |

## A.2   ABLATION STUDY DETAIL

In ablation study of Tables 1,2,3,4,A.3,A.2. The classifier is the official ResNet Pytorch checkpoint, the diffusion model is from Dhariwal & Nichol (2021) and the dataset is ImageNet 128x128. Generating 10000 samples with 250 DDPM steps for evaluation.

Table A.2: Comparative analysis of inputs in guided sampling with respect to ECE. The ResNet is the official Pytorch ResNet checkpoint; the diffusion model is from Dhariwal & Nichol (2021) and the dataset is ImageNet 128x128.

| | $t = 0$ | $t = 20$ | $t = 40$ | $t = 60$ | $t = 80$ |
|---|---|---|---|---|---|
| $\text{ECE}_t \ \hat{x}_t$ | 0.25 | 0.41 | 0.46 | 0.39 | 0.31 |
| $\text{ECE}_t \ \hat{x}_0(t)$ | 0.25 | 0.25 | 0.28 | 0.30 | 0.33 |

Table A.3: Ablation study of sine factor $\gamma$ on classifier guidance with respect to FID. The classifier is the official ResNet Pytorch checkpoint, the diffusion model is from Dhariwal & Nichol (2021) and the dataset is ImageNet 128x128. Generating 10000 samples with 250 DDPM steps for evaluation.

| $\gamma$ | 0.0 | 0.1 | 0.2 | 0.3 | 0.4 |
|---|---|---|---|---|---|
| FID | 5.57 | 5.30 | 5.27 | 5.24 | 5.38 |

## A.3   ABLATION SUMMARY

Table A.4 presents the sequential ablation study of the above designs. The percentage within the brackets is the improvement achieved by each design choice compared to the baseline setting, which

is represented in the first column of each row. The evaluation involves ADM (Dhariwal & Nichol, 2021) generating 10,000 128x128 ImageNet samples using 250 DDPM steps.

Table A.4: Sequential ablation study of the aforementioned designs.

| INPUT TYPES | $\hat{x}_t$ | $\hat{x}_0(t)$ | | | |
|---|---|---|---|---|---|
| FID | 8.61 | **7.17**(16%) | | | |
| SOFTPLUS $\beta$ | $\infty$(RELU) | 5 | 4 | 3 | 2 |
| FID | 7.17 | 6.89 | 6.73 | **6.61**(8%) | 7.02 |
| MARGINAL TEMPERATURE $\tau_2$ | 1.0 | 0.8 | 0.7 | 0.5 | 0.3 |
| FID | 6.61 | 5.62 | 5.45 | **5.27**(20%) | 5.30 |
| SIN FACTOR $\gamma$ | 0.0 | 0.1 | 0.2 | 0.3 | 0.4 |
| FID | 5.27 | 5.30 | 5.27 | **5.24**(1%) | 5.38 |

## B    JOINT VS CONDITIONAL PROBABILITY

Figure B.1 presents the intermediate sampling images and the classifier gradient figures over 250 DDPM steps. Figure B.1 (a) represents the traditional conditional probability settings ($\tau_1 = 1, \tau_2 = 1$): the classifier gradient figure gradually fades from t=50 to 0, indicating a loss of object depiction guidance during the sampling. In contrast, Figure B.1 (b) showcases strengthened joint guidance ($\tau_1 = 1, \tau_2 = 0.5$): the classifier gradient figure increasingly highlights the object outline, providing consistent and accurate guidance direction throughout the entire sampling process.

## C    EXPERIMENT DETAILS

### C.1    OFF-THE-SHELF CLASSIFIER

The off-the-shelf classifiers are the official Pytorch checkpoints at: `https://pytorch.org/vision/main/models/resnet.html`. Specifically, the ResNet50 checkpoint is at: `https://download.pytorch.org/models/resnet50-11ad3fa6.pth`; ResNet101 checkpoint is at: `https://download.pytorch.org/models/resnet101-cd907fc2.pth`.

### C.2    EXPERIMENT ALGORITHM

Algorithm 2 is the EDM off-the-shelf classifier guided sampling.

---

**Algorithm 2** EDM off-the-shelf classifier guided sampling.

---

**Parameter:** SoftPlus activation $\beta$, joint logit temperature $\tau_1$, marginal logit temperature $\tau_2$. classifier guidance scale $\gamma_i$.
**Required:** EDM model $E_\theta$, class label $y$, reverse process sample $\hat{x}_i$, curvate $d_i$, classifier logit of input $x$ on class $y$: $f_y(x)$.
$\hat{x}_N$ sample from $\mathcal{N}(0, \mathbb{I})$
**for** $i \in \{N, ..., 1\}$ **do**
    $d_i, t_i \leftarrow E_\theta(\hat{x}_i)$
    $\bar{x}_i \leftarrow \hat{x}_i / \|\hat{x}_i\|_2$                                                   ▷ sample $\hat{x}_i$ normalization
    $g \leftarrow \nabla_{\bar{x}_i} \log(\exp(f_y(\bar{x}_i)\tau_1)/(\sum_{k=1}^N \exp(f_k(\bar{x}_i)\tau_2)))$
    $\hat{x}_{i-1} \leftarrow \hat{x}_i + (t_i - t_{i-1})d_i + \gamma_i(g/\|g\|_2)$        ▷ normalized classifier gradient as guidance
    **if** $t_{i-1} \neq 0$ **then**
        $\hat{x}_{i-1} \leftarrow E_\theta(\hat{x}_i, \hat{x}_{i-1}, t_i, t_{i-1})$
    **end if**
**end for**
**return** $\hat{x}_0$

---

Algorithm 3 is the DiT off-the-shelf classifier guided sampling.

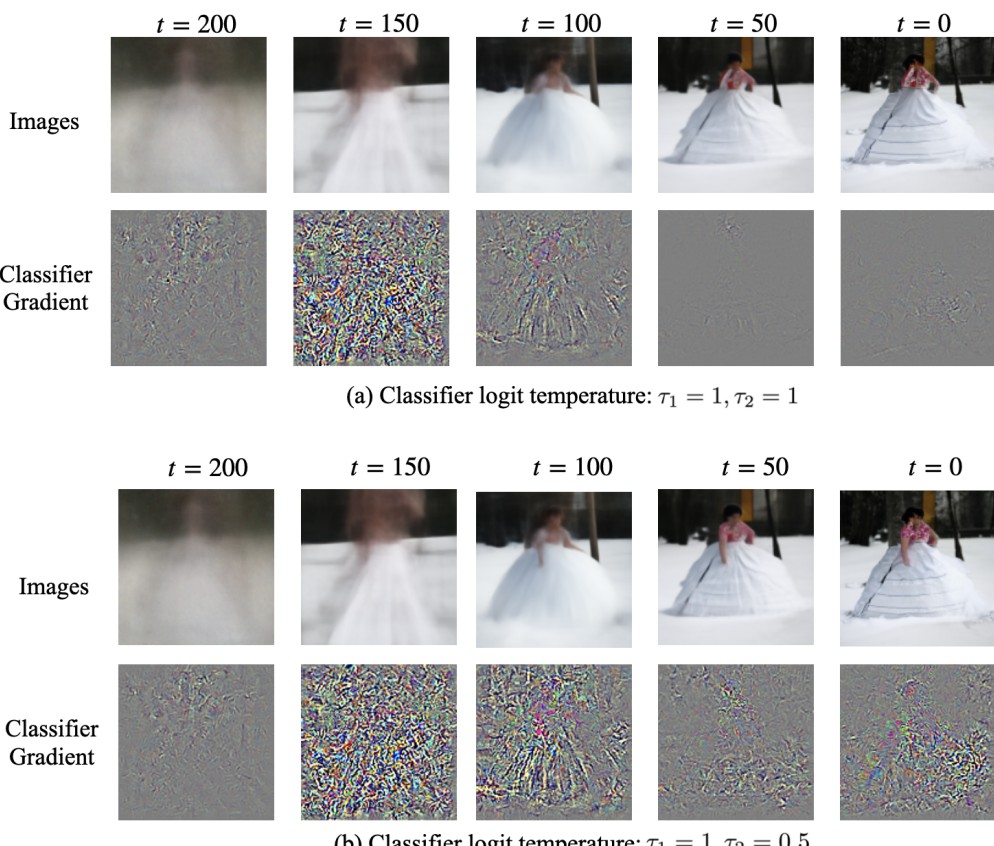

Figure B.1: The illustration of sampling images and classifier gradient figures under conditional probability guidance and joint guidance during the sampling process. The classifier is the official ResNet50 Pytorch checkpoint, the diffusion model is from Dhariwal & Nichol (2021) and the dataset is ImageNet 128x128. The seed is fixed for direct comparison.

## C.3 EXPERIMENT HYPER-PARAMETER

To replicate the DDPM off-the-shelf classifier guided sampling in Table 5:

Softplus $\beta = 3$, joint logit temperature $\tau_1 = 1.0$, marginal logit temperature $\tau_2 = 0.5$, classifier guidance schedule added Sine factor $\gamma_t = 0.3$.

To replicate the EDM off-the-shelf classifier guided sampling in Table 6:

Softplus $\beta = 5$, joint logit temperature $\tau_1 = 1.0$, marginal logit temperature $\tau_2 = 0.0$, classifier guidance schedule added sine factor $\gamma_t = 0.3$. The guidance scale is 0.004 for 10 and 18 sampling steps, 0.0018 for 36 sampling steps, and 0.001 for 256 sampling steps.

To replicate the DiT off-the-shelf classifier guided sampling in Table 7:

classifier-free scale $s = 1.5$, Softplus $\beta = 6$, joint logit temperature $\tau_1 = 1.1$, marginal logit temperature $\tau_2 = 0.5$, classifier guidance schedule added sine factor $\gamma_t = 0.2$,

## C.4 MORE EXPERIMENT

The 256 steps of SDE sampling on ImageNet 64x64 Kingma & Gao (2023) is presented in Table C.1 of Appendix C.4.

---

**Algorithm 3** Off-the-shelf classifier guidance for DiT sampling.

---

**Parameter:** classifier-free scale $s$, classifier guidance scale $\gamma_t$, joint logit temperature $\tau_1$, marginal logit temperature $\tau_2$.

**Required:** DiT model $D_\theta$, VAE decoder $V_D$, classifier logit of input $x$ on class $y$: $f_y(x)$, class label $y$, reverse process class-conditional and unconditional noise $\epsilon_\theta(\hat{z}_t, c, t)$ and $\epsilon_\theta(\hat{z}_t, \emptyset, t)$, reverse process latent and pixel spaced sample $\hat{z}_t$ and $\hat{x}_t$, predicted denoised latent and pixel spaced sample $\hat{z}_0(t)$ and $\hat{x}_0(t)$.

$\hat{z}_T$ sample from $\mathcal{N}(0, \mathbb{I})$

**for** $t \in \{T-1, ..., 0\}$ **do**

$\quad \hat{z}_0(t), \epsilon_\theta(\hat{z}_t, c, t), \epsilon_\theta(\hat{z}_t, \emptyset, t) \leftarrow D_\theta(\hat{z}_t, t)$

$\quad \hat{x}_0(t) \leftarrow V_D(\hat{z}_0(t)) \qquad\qquad \triangleright$ VAE decoder transform latent sample into pixel space

$\quad g \leftarrow \nabla_{\hat{z}_0(t)} \log(\exp(f_y(\hat{x}_0(t))\tau_1)/(\sum_{i=1}^{N} \exp(f_i(\hat{x}_0(t))\tau_2))) \quad \triangleright$ classifier gradient guidance

$\quad \Delta\epsilon_t \leftarrow \epsilon_\theta(\hat{z}_t, c, t) - \epsilon_\theta(\hat{z}_t, \emptyset, t)$

$\quad \bar{g} \leftarrow (g/\|g\|_2)\|\Delta\epsilon_t\|_2 \qquad\qquad\qquad\qquad\qquad \triangleright$ classifier guidance normalization

$\quad \epsilon_t^* \leftarrow \epsilon_\theta(\hat{z}_t, c, t) + (s-1)(\Delta\epsilon_t + \gamma_t \bar{g}) \qquad \triangleright$ classifier guidance into classifier-free

$\quad \mu \leftarrow \frac{1}{\sqrt{\alpha_t}}(\hat{z}_t - \frac{\beta_t}{\sqrt{1-\bar{\alpha}_t}}\epsilon_t^*) \qquad\qquad\qquad \triangleright$ posterior mean Ho et al. (2020)

$\quad \hat{z}_{t-1} \leftarrow$ sample from $\mathcal{N}(\mu, \sigma_t)$

**end for**

$\hat{x}_0 \leftarrow V_D(\hat{z}_0)$

**return** $\hat{x}_0$

---

Table C.1: EDM baseline and the off-the-shelf ResNet guided EDM sampling. Sampled for 256 SDE steps. Generating 50000 ImageNet 64x64 samples for evaluation.

| IMAGENET 64x64 | CLASSIFIER | FID |
|---|---|---|
| EDM BASELINE | - | 1.41 |
| EDM RES101 GUIDED | OFF-THE-SHELF | **1.33** |

## D  CLIP GUIDED FIGURES

The illustration of CLIP-guided diffusion sampling figures. The CLIP is the ViT-L(336px), and the diffusion model Dhariwal & Nichol (2021) is from 256x256 ImagNet.

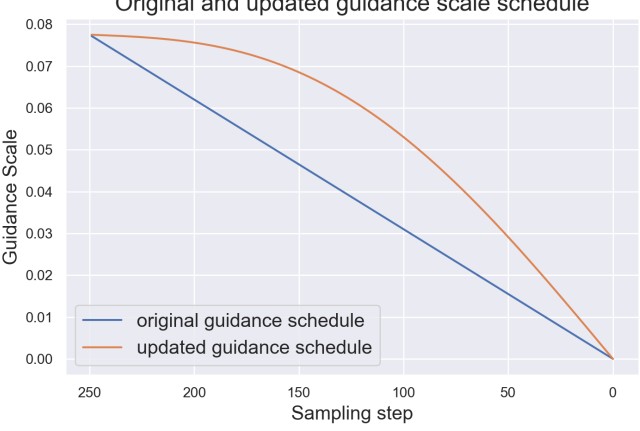

Figure D.1: The comparison of variance schedule and updated variance with $\gamma = 0.3$ schedule.

$$\hat{x}_0(t) = (\hat{x}_t - (\sqrt{1-\alpha_t}\epsilon_t(\hat{x}_t)))/\sqrt{\alpha_t}$$
$$\mu_t(\text{guide}) = \mu_t(\hat{x}_t) + \gamma_t \nabla\text{CLIP}(\hat{x}_0(t), prompt) \tag{D.1}$$

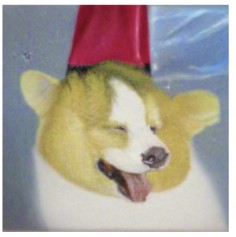 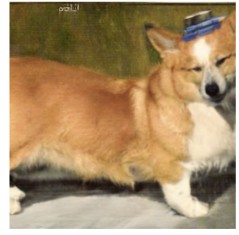 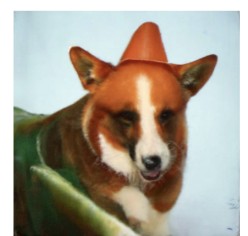 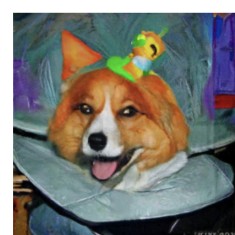

Figure D.2: "Party hat on corgi in oil painting"

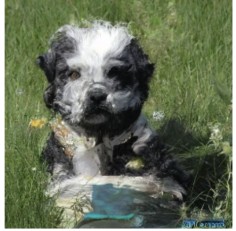 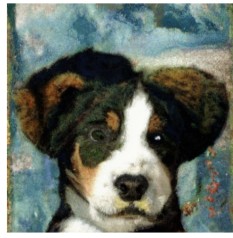 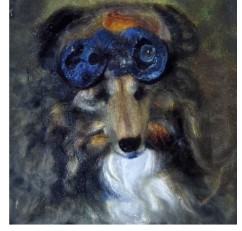 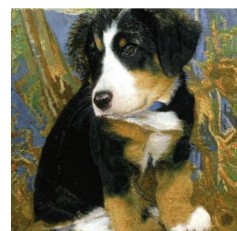

Figure D.3: "Van Gogh Style Dog"

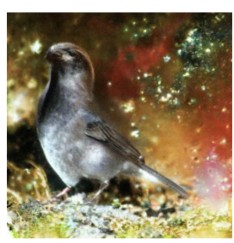 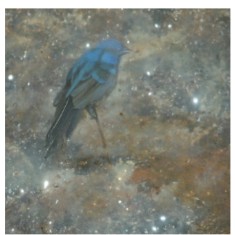 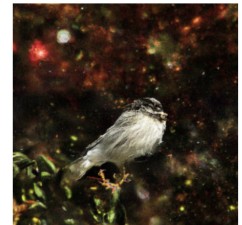 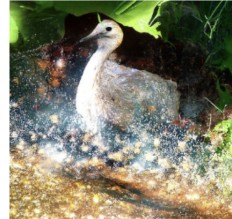

Figure D.4: "Bird in the universe"

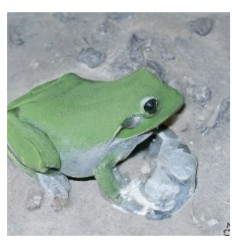 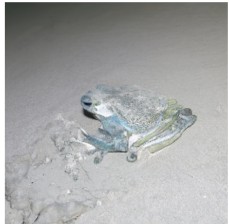 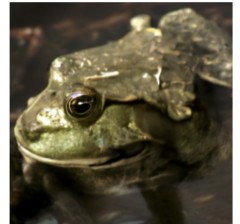 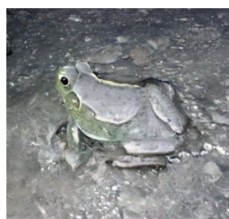

Figure D.5: "Ice Frog"

In Figure D.6, the two CLIP score series are calculated by averaging the text-to-image sampling process using the above prompts "Van Gogh Style Cat, Ice Frog, etc", based on the original linear guidance schedule and our updated guidance schedule respectively. Figure D.6 clearly shows that our updated schedule consistently yields significantly higher CLIP scores throughout the sampling process.

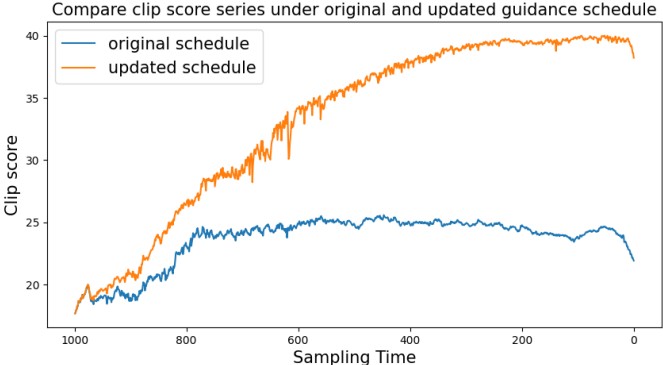

Figure D.6: The comparison of CLIP scores series under linear guidance schedule and updated guidance schedule with sine factor during the sampling process.

## E  PROOF OF PROPOSITION 4.1

This is a direct result of the interpolation inequality (Brezis & Mironescu, 2019). Specifically, the interpolation inequality implies that for any $f \in \mathcal{H}^k(\Omega)$, we have

$$\|\nabla f\|_{L_2(\Omega)} \leq C_1 \|f\|_{\mathcal{H}^k(\Omega)}^{\frac{1}{k}} \|f\|_{L_2(\Omega)}^{1-\frac{1}{k}},$$

and

$$\|f\|_{L_\infty(\Omega)} \leq C_2 \|f\|_{\mathcal{H}^k(\Omega)}^{\frac{d}{2k}} \|f\|_{L_2(\Omega)}^{1-\frac{d}{2k}},$$

where $C_1, C_2$ are constants not depending on $f$, and $d$ is the dimension of the input $\boldsymbol{x}$. Let $\epsilon = p - p_n$, which satisfies

$$\|p - p_n\|_{L_\infty(\Omega)} \leq C_2 C_3 \|\epsilon\|_{\mathcal{H}^k(\Omega)}^{\frac{d}{2k}} \|\epsilon\|_{L_2(\Omega)}^{1-\frac{d}{2k}} = o_{\mathbb{P}}(1). \tag{E.1}$$

Also, we have

$$\|\nabla p - \nabla p_n\|_{L_2(\Omega)} \leq C_1 \|\epsilon\|_{\mathcal{H}^k(\Omega)}^{\frac{1}{k}} \|\epsilon\|_{L_2(\Omega)}^{1-\frac{1}{k}} = o_{\mathbb{P}}(1). \tag{E.2}$$

Thus, it can be shown that

$$\begin{aligned}
\|\nabla \log p - \nabla \log p_n\|_{L_2(\Omega)} &= \left\| \frac{\nabla p}{p} - \frac{\nabla p_n}{p_n} \right\|_{L_2(\Omega)} \\
&= \left\| \frac{p_n \nabla p - p \nabla p_n}{p(p - \epsilon)} \right\|_{L_2(\Omega)} \\
&\leq \frac{\|p_n - p\|_{L_2(\Omega)} \|\nabla p\|_{L_2(\Omega)} + \|p\|_{L_2(\Omega)} \|\nabla p - \nabla p_n\|_{L_2(\Omega)}}{\|p\|_{L_\infty(\Omega)} (c_1 - \|\epsilon\|_{L_\infty(\Omega)})} \\
&= o_{\mathbb{P}}(1),
\end{aligned}$$

where the last equality is because the Sobolev embedding theorem implies $\|\nabla p\|_{L_2(\Omega)} \leq C_4 \|p\|_{\mathcal{H}^k(\Omega)}$, and by Eq.E.1 and Eq.E.2. This finishes the proof.

## F PROOF OF PROPOSITION 4.2

For notational simplicity, let $h_l(x) = P(Z = l | X = x)$ and $g_l(x) = P(Z = l, X = x)$. Taking the gradient with respect to $x$, it can be seen that

$$
\begin{aligned}
\nabla h_l(x) =& \nabla \left( \frac{g_l(x)}{f(x)} \right) \\
&\propto \frac{b_l f_l(x) \Sigma_l^{-1}(\mu_l - x) f(x) - b_l f_l(x)(\sum_{k=1}^K b_k e^{-\frac{1}{2}(x - \mu_k)^\top \Sigma_k^{-1}(x - \mu_k)} \Sigma_k^{-1}(\mu_k - x))}{f(x)^2} \\
&\propto \Sigma_l^{-1}(x - \mu_l) f(x) - \sum_{k=1}^K b_k e^{-\frac{1}{2}(x - \mu_k)^\top \Sigma_k^{-1}(x - \mu_k)} \Sigma_k^{-1}(\mu_k - x) \\
&\propto \sum_{k=1}^K b_k e^{-\frac{1}{2}(x - \mu_k)^\top \Sigma_k^{-1}(x - \mu_k)} (\Sigma_l^{-1}(x - \mu_l) - \Sigma_k^{-1}(x - \mu_k)).
\end{aligned}
$$

Direct computation shows that

$$
\nabla g_l(x) = b_l f_l(x) \Sigma_l^{-1}(\mu_l - x) \propto \Sigma_l^{-1}(\mu_l - x).
$$