# OpenReview forum: "Elucidating the design space of classifier-guided diffusion generation"
_ICLR.cc/2024/Conference — ICLR 2024 poster_

### Official Review · Reviewer_VZ9L · 2023-10-29

**Soundness:** 3 good
**Presentation:** 3 good
**Contribution:** 3 good
**Rating:** 5
**Confidence:** 5

**Summary:**

The paper focuses on a very interesting question of guiding the diffusion modes by leveraging off-the-shelf classifiers in a training-free fashion. Specifically, the authors first provide a simple analysis to demonstrate that the off-the-shelf classification can achieve better accuracy than the fine-tuned classifier when the noise level is high, which may have been ignored in the previous works. Then, the authors turn to exploit the pre-trained classifier for guiding diffusion generation with comprehensive consideration of the detailed settings.

**Strengths:**

1.	In my opinion, the major contribution of this paper lies in Section 4.1, which provides an empirical analysis by evaluating the calibration of both fine-tuned and off-the-shelf classifiers.  The results reveal that fine-tuned classifiers are less calibrated than off-the-shelf ones when the noise level is high, which indicates that Off-the-shelf classifiers’ potential is far from realized.
2.	To optimize the design of the proposed methods, the authors have considered multiple aspects, including the classifier inputs, smoothing guidance, guidance direction, and guidance direction, and designed the corresponding strategies to improve the overall performance.

**Weaknesses:**

1.	 The paper focuses on the classic class-conditional diffusion generation, which is a simple case in the text-to-image generation. How about the performance of the proposed methods for the general case with text prompts as conditions? In particular, in section 5.4, the authors also provide a simple analysis of this case with the CLIP model. However, the results are not good.
2.	Human-level metrics should be involved for clearer comparisons. It is well-known that some quantitative metrics like FID, may be problematic in some cases. In particular, the FID metric, used in ablation analysis in section 4 and experiments in section 5, cannot measure conditional adherence which is important for conditional generation. CLIP score should also be considered to evaluate the performance.
3.	It is weird that the final images are merely the same as each other in Figure 4 with different settings of logit temperature, while the FID score varies with different settings in Table 4.

**Questions:**

My major commons lie in the experimental evaluation. Please provide more experimental analysis or discussions to validate the effectiveness and robustness of the proposed methods.

How about the performance in terms of CLIP score and Human-level metrics?

How about the performance for the conditional generation with the general text prompts?

Please provide more discussions about Table 4 and Figure 4.

---

> ### Author Response · Authors · 2023-11-19
> **Response to Reviewer VZ9L**
>
> Thanks for the helpful comments. Below, we address the raised concerns.
>
>
> ## How about the performance of the proposed methods for the general case with text prompts as conditions? In particular, in section 5.4, the authors also provide a simple analysis of this case with the CLIP model. However, the results are not good.
>
> Thanks for the question. We agree that the more general text prompt condition is very important. The class-conditional setting, although simple, does provide a solid test bed for different kinds of guidance methods, with well-accepted quantitative measurements.
> If one method is not good at class-guided generation, we wouldn't expect it to excel at the more complicated text prompts.
> Therefore, we think our evaluations on the more typical class-guided generation case are valuable and the findings can potentially be generalized to more general settings.
>
>
> Nevertheless, we have explored the CLIP-guided generation task.
> In the added Figure 6 of our revised paper, we present a comparison of CLIP scores between the original guidance schedule and our proposed schedule. Notably, our updated schedule consistently achieves significantly higher CLIP scores throughout the sampling process. The objective of our CLIP-guided sampling approach is to showcase the capability of a traditional off-the-shelf CLIP model in guiding the ImageNet-trained diffusion model (ADM [Dhariwal et al. 2021]). By leveraging this guidance, our model generates images that surpass the limitations of the ImageNet dataset and align with the provided prompt.
>
> We aim to investigate more complicated discriminative models such as BLIP-VQA, as well as discriminative models in other modalities as guidance in the future.
>
>
> ## Human-level metrics should be involved for clearer comparisons... CLIP score should also be considered to evaluate the performance.
>
> Given the subjectivity of human-level metrics, we primarily focus on utilizing quantitative metrics to evaluate the performance of the classifier-guided diffusion (including our extensive experiments in evaluating the classifier-guided diffusion with quantitative metrics across diverse diffusion architecture: DDPM, EDM, DiT). For a more comprehensive evaluation, we add more quantitative metrics to the experiments. There are some values missing because the authors did not provide them.
>
>
> Table 1: ImageNet 64x64 evaluation: 250 steps of DDPM sampling results.
> |                    |   IS   |   FID  |   sFID  | Precision | Recall
> |---|---|---|---|---|---
> | ADM                |   -    |  2.61  | 3.77  | 0.73  |  0.63
> | ADM-G              |   -    |  2.07  | 4.29  | 0.74  | 0.63
> | Resnet101-guided(Ours)   |  70.52 |  1.72  | 3.36  | 0.76  | 0.60
>
> Table 2: ImageNet 128x128 evaluation: 250 steps of DDPM sampling results.
> |                 |   IS   |   FID  |  sFID | Precision| Recall
> |---|---|---|---|---|---
> | ADM             |    -   |  5.91  | 5.09  | 0.70  | 0.65
> | ADM-G           | 182.69 |  2.98  | 5.09  | 0.78  | 0.59
> | CFG             | 158.47 |  2.43  | -     |   -   |  -
> | Resnet101-guided(Ours)| 187.83 | 2.19   | 4.53  | 0.79 | 0.58
>
> Table 3: ImageNet 128x128 evaluation: 25 steps of DDIM sampling results.
> |                 |   IS   |   FID  |  sFID | Precision| Recall
> |---|---|---|---|---|---
> | ADM-G           |   -    |  5.98  | 7.04  | 0.78  | 0.51
> | Resnet101-guided(Ours)|  139.88|  5.72  |  5.56 | 0.73  |  0.55
>
>
> In the added Figure 6 of our revised paper, we present a comparison of CLIP scores between the original guidance schedule and our proposed schedule. Notably, our updated schedule consistently yields significantly higher CLIP scores throughout the sampling process.
>
>
> ## It is weird that the final images are merely the same as each other in Figure 4 with different settings of logit temperature, while the FID score varies with different settings in Table 4.
>
> In Figure 4 of our paper, the random seed was fixed to ensure a consistent and strict comparison, thereby minimizing dramatic changes. However, it is worth noting that some significant differences still exist between the two settings. For example, in the left-hand side figure representing the traditional guidance setting, the generated dog gradually loses the depiction of its feet from time 50 to 0. In contrast, in the right-hand side figure where the joint guidance was amplified, the dog's feet remain clear, and there is a noticeable enrichment of environmental details.

---

> > ### Comment · Reviewer_VZ9L · 2023-11-21
> > **Thanks for your responses.**
> >
> > We sincerely appreciate the authors' helpful responses, which have effectively addressed most of my inquiries, particularly regarding the CLIP score metric. However, I prefer to hold my score at this time, due to the concerns on the generalizability of the proposed method. Additionally, I believe that the paper would greatly benefit from an extension of the current method to encompass the broader task of text-to-image synthesis.

---

### Official Review · Reviewer_wqgP · 2023-11-01

**Soundness:** 3 good
**Presentation:** 3 good
**Contribution:** 3 good
**Rating:** 6
**Confidence:** 5

**Summary:**

The manuscript analyzes the limitations of current classifier guidance in terms of flexibility, calibration error as well as smoothness. Based on the analysis, the authors propose to use an off-the-shelf classifier instead of a noise-aware classifier for guidance which shows some encouraging results.

**Strengths:**

The problem is very interesting, the method is simple yet effective. The current approach requires finetuning classifiers as well as training diffusion models to be a joint model. This work removes this limitation and shows that the off-the-shelf classifier has the potential to outperform noise-aware models.

**Weaknesses:**

Although the work is very interesting, there are still some questions:

1. The comparison is not so comprehensive. Since the main baseline should be the noise-aware classifier, the work provides little comparison in Table 5. More metrics such as IS, sFID, Precision and Recall should be included.
2. In Table 5, results for IMN64x64 as well as IMN256x256 for DDPM from Dhariwal and Nichol (2021) is missing. Whether or not the method can perform well on other resolutions.
3. Does off-the-shelf classifier guidance provide conditional information for the unconditional diffusion model? In my belief, the main objective of the guidance should be providing conditional information for the unconditional diffusion model rather than just improving the generated image quality. In Table 5, the Diffusion model on ImageNet128x128 is conditionally trained. Thus, the conditional information from an off-the-shelf classifier is not important. Results for combining off-the-shelf classifiers with unconditional models should be included.
4. How off-the-shelf models Resnet can be used with ImageNet64x64 or ImageNet128x128 although these models are trained on ImageNet224x224? Besides, the generated images are clip to be in range of [-1; 1], how does it fit with model trained with input images normalized by ImageNet datasets?
5. There is a fatal gap in the modeling part in Algorithm 1, in the sampling equation $\hat{x}_{t-1} \sim \mathcal{N}(\mu + \gamma _t g, \sigma _t)$. In the original paper by Dhariwal and Nichol (2021), the sampling resulted from the $\log(p _{ \theta }(x_t|x _{t+1}) p _{\phi}(y|x_t))$. However, given the structure of Algorithm1, this equation is no longer valid since the gradient is taken regarding $x _0 (t)$ instead of $x_t$. In order to apply the same sampling process, even when $x_0(t)$ is forward through the classifier, the gradient should be taken regarding $x_t$.
6. I guess from the point (5), this is the main reason why the method can not be applied to DDIM?
7. Ablation study is missing, it is quite vague to understand which proposed scheme is the main course for the improvement. From my understanding, there are three main differences from normal classifier guidance which are:
* Gradients via $x_0(t)$ instead of $x_t$
* Guidance schedule
* $\tau _2$ temperature

However, discussion on the contribution of each of them is missing

8. Given three contributions as in (7), does the performance of the noise-aware classifier guidance also get improvements?
9. Besides Resnet, how are other architectures such as DenseNet, Transformers? Do they also work with this scheme?
10. CLIP guidance should be compared against the noise-aware CLIP guidance
11. The connection from 4.1, 4.2, 4.3 and 4.4, 4.5 as well as the design of the algorithm lacks some connections.

It seems that the paper is written in hurry so that the format of the paper is not really good as well as some errors in equations:
1. Equation (2) should be $\log exp(\tau f _{y(x)})$? Check format of the equation (2) also.
2. Equation (3) should be $\log exp(\tau _1 f _{y(x)})$?

**Questions:**

See the weaknesses. The work is interesting and potential, yet there are a number of concerns as well as writting. Will raise the score if all concerns are solved.

**Details Of Ethics Concerns:**

N.A

---

> ### Author Response · Authors · 2023-11-19
> **Response to Reviewer wqgP (Part 1)**
>
> Thanks for the helpful comments. Below, we address the raised concerns.
>
>
> ## "The comparison is not so comprehensive. Since the main baseline should be the noise-aware classifier, the work provides little comparison in Table 5. More metrics such as IS, sFID, Precision and Recall should be included."
>
> In our research, we aim to demonstrate the potential of the more accessible traditional ResNet classifier and prove that even without specifically noise-finetuned, the off-the-shelf classifier can guide the diffusion to generate high-quality sampling: FID: 2.19 in ImageNet128x128) not only largely surpass the noise-aware fine-tuned classifier guided diffusion [Dhariwal et al. 2021] (FID: 2.97) but also the classifier-free trained diffusion (FID: 2.43) [Ho et al. 2022].
>
> Due to the computation resource and time limits, we prioritize testing the ImageNet64x64 setting and we plan to include the ImageNet 256x256 results in our future research to provide a comprehensive analysis across different image resolutions. There are some values missing because the authors did not provide them.
>
> Table 4: ImageNet 64x64 evaluation: 250 steps of DDPM sampling results.
> |                    |   IS   |   FID  |   sFID  | Precision | Recall |
> |---|---|---|---|---|---
> | ADM                |   -    |  2.61  | 3.77  | 0.73  |  0.63
> | ADM-G              |   -    |  2.07  | 4.29  | 0.74  | 0.63
> | Resnet101-guided (Ours)   |  70.52 |  1.72  | 3.36  | 0.76  | 0.60
>
> Table 5: ImageNet 128x128 evaluation: 250 steps of DDPM sampling results.
> |                 |   IS   |   FID  |  sFID | Precision| Recall
> |---|---|---|---|---|---
> | ADM             |    -   |  5.91  | 5.09  | 0.70  | 0.65
> | ADM-G           | 182.69 |  2.97  | 5.09  | 0.78  | 0.59
> | Resnet101-guided (Ours) | 187.83 | 2.19   | 4.53  | 0.79 | 0.58
>
>
> ## Resolution: "In Table 5, results for IMN64x64 as well as IMN256x256 for DDPM from Dhariwal and Nichol (2021) is missing... How off-the-shelf models Resnet can be used with ImageNet64x64 or ImageNet128x128 although these models are trained on ImageNet224x224? Besides, the generated images are clip to be in range of [-1; 1], how does it fit with model trained with input images normalized by ImageNet datasets?"
>
> Inside the ResNet, there is an average layer: torch.nn.AdaptiveAvgPool2d((1,1)), which is capable of reshaping the different pixel input into the fixed size.
> During guided sampling, we primarily adhere to the guidance definition outlined in ADM [Dhariwal et al. 2021]. In this approach, a gradient is computed with respect to the generated samples since the guidance is incorporated into the generated images. Additionally, the guidance is computed as the gradient of the logarithm of softmax classifier logits, which is then normalized through an exponential operation across all classes.
>
>
> ## "Does off-the-shelf classifier guidance provide conditional information for the unconditional diffusion model? Classifier guidance vs. conditional diffusion. "
>
> According to the findings presented in Table 4 of ADM [Dhariwal et al. 2021], classifier-guided unconditional diffusion falls short in terms of performance (FID: 12.0) compared to classifier-guided conditional diffusion (FID: 4.59). Consequently, the subsequent experiments in Table 5 of ADM [Dhariwal et al. 2021] specifically focus on classifier-guided conditional diffusion.
> To ensure a fair comparison with SOTA results, we have adopted the guided conditional diffusion setting in this study.
>
> Additionally, DiT is trained in a classifier-free manner, which encompasses both class-conditional and unconditional conditions. Table 9 of our revised paper demonstrates that off-the-shelf classifier guidance can also be employed in classifier-free diffusion.
> However, we acknowledge that there are further settings and configurations to explore, which will be addressed in future research endeavors.

---

> > ### Author Response · Authors · 2023-11-19
> > **Response to Reviewer wqgP (Part 2)**
> >
> > ## "There is a fatal gap in the modeling part in Algorithm 1....this equation is no longer valid since the gradient is taken regarding ... this is the main reason why the method can not be applied to DDIM?"
> >
> > Here, $\hat{x}_0(t)$ refers to the denoised sample $\hat{x}_t$, denoted as $\hat{x}_0(t)$ = $(\hat{x}_t - (\sqrt{1-\alpha_t})$ $\epsilon \theta(\hat{x}_t,t))$ $ / \sqrt{\alpha_t}$. Ideally, the classifier gradient guides the diffusion denoising process towards the target image. The sampling trajectory from  $\hat{x}_0(T)$ to $\hat{x}_0(0)$ may not be fundamentally different from that from $\hat{x}_T$ to $\hat{x}_0$.
> >
> > Practically, we think that for off-the-shelf classifiers, choosing denoised $\hat{x}_0(t)$ as inputs may be more compatible since they are not specifically fine-tuned on noisy data.
> > This choice is used in [Dhariwal et al. 2021] but the quantitative comparison between $\hat{x}_0(t)$ and $x_t$ is missing.
> > In Table 2 of our revised paper, we select $\hat{x}_0(t)$ as the input to the classifier due to its superior guidance performance.
> > In the ablation study table of classifier input types, we generate 10,000 samples from the 128x128 ImageNet dataset using ADM [Dhariwal et al. 2021], utilizing 250 DDPM steps.
> >
> > Table: Classifier input types ablation study:
> > | Inputs |   $\hat{x}_t$   | $\hat{x}_0(t)$
> > |---|---|---
> > | FID    |  8.61  | 7.17(16\%)
> >
> >
> > To provide clarification of the off-the-shelf guidance in the DDIM sampling setting, we have included the 25 steps of DDIM guided-sampling experiments in the table below, it demonstrates that our off-the-shelf classifier guidance also outperforms the fine-tuned classifier-guided ADM-G [Dhariwal et al. 2021] in DDIM.
> >
> > Table 6: ImageNet 128x128 evaluation: 25 steps of DDIM sampling results.
> > |                 |   IS   |   FID  |  sFID | Precision| Recall
> > |---|---|---|---|---|---
> > | ADM-G           |   -    |  5.98  | 7.04  | 0.78  | 0.51
> > | Resnet101-guided (Ours) |  139.88|  5.72  |  5.56 | 0.73  |  0.55
> >
> >
> > ## "Ablation study is missing"
> > Thanks for the question. From Section 4.2 to Section 4.5, we have provided ablation studies for each of the design choices. We agree that it's important to look at all of them together in one task. Hence, we have provided additional ablation experiments.
> >
> > The table below (Table 6 of our revised paper) showcases the results of our off-the-shelf classifier guided sampling in sequential ablation experiments, where we explore different design choices. The sequential parameter rule is employed to identify the selection that yields the best FID value. The percentage number within the brackets of each row indicates the improvement achieved by each design choice compared to the baseline setting, which is represented in the first column of each row. The evaluation process involves generating 10,000 samples from the 128x128 ImageNet dataset using ADM[Dhariwal et al. 2021], utilizing 250 DDPM steps.
> >
> > Table: Sequential ablation:
> > | Inputs |   $\hat{x}_t$   | **$\hat{x}_0(t)$** |  |  |   |
> > |---|---|---|---|---|---
> > | FID    |  8.61  | **7.17 (16\%)**    |   |   |
> > | Softplus $\beta$ | inf (ReLU) |  5.0 | 4.0 | **3.0** |  2.0 |
> > | FID              | 7.17 |  6.89  |  6.73  |  **6.61 (8\%)** | 7.02  |
> > | marginal temperature $\tau_2$| 1.0 | 0.8 | 0.7 | **0.5** | 0.3 |
> > | FID                          | 6.61 |  5.62  | 5.45 | **5.27 (20\%)** | 5.30 |
> > | sin factor $\gamma$ | 0.0 | 0.1 | 0.2 | **0.3** | 0.4 |
> > | FID                 | 5.27 | 5.30  | 5.27 | **5.24 (1\%)** | 5.38 |
> >
> >
> > ##  Does the performance of the noise-aware classifier guidance also get improvements? CLIP guidance should be compared against the noise-aware CLIP guidance
> >
> >
> > Our objective with CLIP-guided sampling is to illustrate that a standard off-the-shelf CLIP model can effectively guide the diffusion model (trained solely on the ImageNet dataset) and generate visually appealing text-to-image results. In contrast, GLIDE [Nichol et al. 2021] necessitates additional training on carefully chosen noisy samples, which introduces extra complexity. In Figure 6 of our revised paper, we compare the CLIP scores achieved with the original guidance schedule and our proposed schedule. Our updated schedule consistently produces considerably higher CLIP scores throughout the sampling process, demonstrating the efficacy of our approach.
> >
> > ## Besides Resnet, how are other architectures such as DenseNet, Transformers? Do they also work with this scheme?
> >
> > In this research, our primary focus lies on the classical ResNet classifier, given its prominence as a fundamental architecture in classification models. We apply this classifier to various diffusion models, including DDPM, EDM, and DiT. However, we acknowledge the potential for further exploration of different classifier architectures, which we intend to pursue in future research endeavors.

---

> > > ### Author Response · Authors · 2023-11-19
> > > **Response to Reviewer wqgP (Part 3)**
> > >
> > > ## Writing: The connection from 4.1, 4.2, 4.3 and 4.4, 4.5 as well as the design of the algorithm lacks some connections.
> > >
> > > The integral calibration error, denoted as $\overline{\text{ECE}}$, plays a crucial role in our research. It assesses the calibration of the off-the-shelf classifier throughout the diffusion process. Proposition 4.1 establishes a connection between classifier guidance and probability estimation, emphasizing the significance of achieving a lower calibration error. In Figure 1, we analyze $\overline{\text{ECE}}$ values and demonstrate that the off-the-shelf ResNet achieves lower $\text{ECE}_t$ values than the fine-tuned classifier when the noises are large. This finding provides a strong basis for asserting that the off-the-shelf classifier has the potential to match the performance of fine-tuned classifier guidance.
> > >
> > > To leverage this potential, we utilize the predicted denoised image $\hat{x}_0(t)$ as the input to the classifier, as suggested in universal guidance [Bansal et al. 2023]. To achieve improved calibration, we propose three key design elements: smooth classifier guidance, consistent guidance direction, and an updated guidance schedule. These design choices aim to calibrate the off-the-shelf classifier with the diffusion models and enable the generation of high-quality guided samples. By incorporating these elements, we enhance the calibration of the off-the-shelf classifier and facilitate the production of superior samples.
> > >
> > > We have refined the writing at the end of Section 4.1 to better transition to the sections afterward.
> > >
> > > ## errors in equations [2] and [3]
> > > In equations (2) and (3) of the paper, you are absolutely correct that the numerator terms of the guidance are $\log{\exp(\tau f_y(x))}$ and $\log{\exp(\tau_1 f_y(x))}$. We simplify these terms by taking advantage of the fact that the exponential function is already contained within the logarithmic operation. Thank you for pointing it out.

---

> ### Author Response · Authors · 2023-11-22
> **Appreciation**
>
> Dear Reviewer wqgP,
>
> Thank you for your detailed review and valuable feedback.
> We sincerely hope our previous response has addressed your questions and concerns.
> We look forward to hearing back from you and addressing any further questions or concerns you may have.
> Thank you for your time.

---

### Official Review · Reviewer_rssc · 2023-11-03

**Soundness:** 2 fair
**Presentation:** 2 fair
**Contribution:** 2 fair
**Rating:** 5
**Confidence:** 4

**Summary:**

The paper presents a novel method to improve guidance in conditional diffusion generation without additional training by utilizing off-the-shelf classifiers. The authors introduce pre-conditioning techniques that enhance the performance of existing state-of-the-art diffusion models by up to 20% on ImageNet. Their approach is efficient, scalable, and leverages the widespread availability of pretrained classifiers, promising advancements especially in text-to-image generation tasks.

**Strengths:**

The paper adeptly integrates elements from both classifier-free and classifier-based diffusion approaches. Additionally, it employs off-the-shelf classifiers to enhance performance while maintaining efficiency. The experimental results presented confirm the effectiveness of these methods.

**Weaknesses:**

1. Sec 4.2 "PREDICTED DENOISED SAMPLES" seems trivial. This is already explored by the CVPR paper " Universal guidance for diffusion models". The authors should not make this part a separate subsection if this is not the author's original work.

2. Sec 4.3 "SMOOTH CLASSIFIER" seems trivial. I think there are already many works which use Softplus as activation function and explore its difference/advantage with ReLU. If I understand it correctly, the only contribution here is to replace ReLU with Softplus. The novelty point is not enough for an ICLR paper.

3. Sec 4.4 "JOINT VS CONDITIONAL DIRECTION", the author mention "we reduce the value of marginal temperature", which seems kind of manual tuning parameters. Is there some validation metric the authors use to determine the optimal temperature?

4. Sec 4.5 "GUIDANCE SCHEDULE" seems trivial. If I understand it correctly, the only contribution here is introducing a sin factor. This seems more like a trick instead of some research contribution. For it to be a research contribution, the author should first discuss what kind of guidance is good and why the author choose the sin factor here.

5. Sec 5.3 "OFF-THE-SHELF GUIDANCE FOR DIT", the authors propose to incorporate classifier guidance g into classifier-free sampling. The idea is straightforward but the same question the authors introduce a parameter gamma_t here. How do the authors choose a proper gamma_t for a specific case?

**Questions:**

See Weaknesses.

---

> ### Author Response · Authors · 2023-11-19
> **Response to Reviewer rssc**
>
> Thanks for your time in reviewing our work.
> Below, we address the raised concerns.
> We hope you can give our work the benefit of the doubt and look at it as a whole, rather than singling out each of its elements.
>
> ## Sec 4.2 "PREDICTED DENOISED SAMPLES" seems trivial.
>
> The idea of using predicted denoised samples is not hard to think of and it has already been applied by other methods, e.g., universal guidance [Bansal et al. 2023]. However, why it is a better choice still lacks justification.
> We do not claim the proposal of it as one of our novelties, but we did offer additional empirical results including integral calibration error $\overline{\text{ECE}}$ and FID as justification, as shown in Table 2 of our paper. Such a detailed comparison has not been made in previous work.
>
> ## Sec 4.3 "SMOOTH CLASSIFIER" seems trivial.
>
> The smoothness of the classifier is linked to its calibration. This may seem trivial but to the best of our knowledge, we are the first to point out calibration as one of the guidelines for classifier-guided diffusion generation.
> Further, we take the diffusion process into consideration and propose to look at the expected calibration error along the diffusion process.
>
> In Figure 1 of our paper, surprisingly, we found that off-the-shelf classifiers are more calibrated than the specifically fine-tuned ones when the noise levels are high.
> We followed up with this observation with guided generation experiments.
> In Appendix A.1, we found that the integral calibration error is strongly positively correlated with the performance
> This has been one of our biggest motivations for improving the classifier calibration.
> We have extended this part in Section 4.1 to make a more convincing case.
>
> On the practical side, we propose to augment the activation function for calibration, from ReLU to softplus, and choose the hyperparameter $\beta$ by examining the ECE error.
> This type of modification has been explored in [Zhu et al. 2021] for improving adversarial transferability but not for guiding diffusion generation.
> With this design choice, we are able to achieve better performance than well-established classifier guidance and classifier-free guidance on ImageNet benchmarks, which is not known to be possible before.
>
> ## Sec 4.4 "JOINT VS CONDITIONAL DIRECTION", the author mention "we reduce the value of marginal temperature", which seems kind of manual tuning parameters. Is there some validation metric the authors use to determine the optimal temperature?
>
> Empirically, we determine the hyperparameter for the marginal temperature based on a lightweight guided sampling of 1000 generated samples. However, we emphasize that the true value lies in the underlying insight. This is supported by the ablation study presented in Table 4 of our paper (we put it below), where all reduced values of the marginal temperature exhibit significant improvements over the baseline($\tau_2=1$), differing only in magnitude.
>
>
> Table: Ablation study of marginal logit temperature $\tau_2$ with respect to FID. $\tau_1$ is fixed as 1. The evaluation process involves generating 10,000 samples from the 128x128 ImageNet dataset using off-the-shelf classifier-guided ADM[4], utilizing 250 DDPM steps.
> |      Marginal temperature           |   1.0 (base)   |   0.8  |  0.7 |  0.5  | 0.3
> |---|---|---|---|---|---
> | FID             |    6.20   | 5.62  | 5.45  | 5.27 | 5.30
>
>
>
> ## Sec 4.5 "GUIDANCE SCHEDULE" seems trivial.
>
> The inspiration for the proposed new guidance schedule comes from the observation of CLIP scores' evolution in the CLIP-guided experiment, as depicted in the added Figure 6 of our revised paper. We observed that the CLIP scores only increased at the beginning of the guided sampling process (from 1000 to 800 time steps), but plateaued afterward. Consequently, we introduced a sine function to modify the original guidance schedule. This choice was motivated by two reasons: firstly, the sine function is continuous and smooth, allowing for a seamless transition in guidance; secondly, it amplifies the guidance, particularly during the early to middle stages of the guided sampling process, shown in Figure D.1 of the appendix.
>
> ## Sec 5.3 "OFF-THE-SHELF GUIDANCE FOR DIT", how do the authors choose a proper gamma_t for a specific case?
>
> When it comes to selecting the appropriate choices for $\gamma_t$, we typically employ a lightweight guided sampling of 1000 generated samples for evaluation.

---

> > ### Comment · Reviewer_rssc · 2023-11-21
> > **Thanks for the feedback**
> >
> > Thanks for the feedback. It did clarify some of my concerns. I think the overall novelty is not enough as it involves many manual design.
> >
> > I have increased my score to 5.

---

### Official Review · Reviewer_r6Zr · 2023-11-04

**Soundness:** 2 fair
**Presentation:** 2 fair
**Contribution:** 3 good
**Rating:** 6
**Confidence:** 3

**Summary:**

In this paper, the authors analyze off-the-shelf classifier guidance diffusion through multiple perspectives including calibration, smoothness, score decomposition and guidance scale scheduling. They use calibration error as a new metric to assess the performance of classifier guidance diffusion and propose several techniques to improve this task. They conduct experiments on multiple diffusion models with pre-trained ResNet classifiers and show consistent improvement with their proposed method compared to baselines.

**Strengths:**

This paper provides several interesting techniques to improve off-the-shelf classifier guidance diffusion. These techniques are practical, training-free and fairly easy to be incorporated into the current diffusion frameworks. Their analysis also provides interesting insight into what happens in the process of classifier guidance diffusion from multiple different perspectives.

**Weaknesses:**

1. I don’t really see the direct connection between Proposition 4.1 and Eq. 1. There seems to be a gap between ECE and $\|p_n-p\|$. Also to make the paper self contained, “bins”, “acc”, “conf” should be defined before mentioning.

2. It is unclear to me how Proposition 1 and ECE inform the design choices of the method. For Section 4.2, the same conclusion can be drawn with only FID. For Section 4.3, Proposition 1 only says “with smoothness $k>1$”, but it doesn’t claim better calibration/score prediction with higher smoothness. Section 4.4 and 4.5 seems to be completely irrelevant to ECE.

3. Relating to Weakness 2, I think the story line of this paper is a little bit scattered. There are many components in the story and it is difficult to tell which one is the main contribution of this paper. The components can also be better connected.

4. There are four components to the proposed method: (1) use $ \nabla_{\hat{x_0}(t)} \log p(y|\hat{x_0}(t))$ instead of $\nabla_{\hat{x_t}} \log p(y|\hat{x_t})$ (2) use Softplus activation to increase the smoothness (3) use a second temperature to adjust the “ratio” of joint and marginal guidance (4) sin factor guidance schedule. Since there is no ablation study that involves all four components conducted, it is very difficult to tell which one is actually effective. It would be great if the authors can include experiments that gradually exclude these components one-by-one to see which one is the most effective.

Minor suggestions:

1. According to the official style files provided by ICLR call for papers, the appendix should be included in the same PDF file as the main text and the bibliography.

2. Eq. 1 $k$ notation conflicts with the smoothness $k$ in Proposition 4.1.

**Questions:**

1. The formula for $\text{ECE}_t$ is with respect to $\hat{x_t}$ but in Section 4.2 the authors talked about using $\hat{x_0}(t)$ will provide better calibration, so which sample did the authors use when providing the ECE results for the rest of the experiments?

2. It is unclear to me how did the authors incorporate the Softplus activation function into the pre-trained classifiers, did they just replace all the activation functions in the pre-trained models? Or is there anything else that they did?

3. How did the authors calculate the marginal likelihood for CLIP guidance generation?

4. What is the recurrent step in Table 1? Is it the same as the “backward guidance” in “Universal Guidance for Diffusion Model” paper? And why is the inference time not changed significantly with the calibrated method given there is extra marginalization required for all classes?
What type of GPU did the authors use in their experiments?

I am happy to raise my score if the authors address my concerns and questions in the rebuttal.

---

> ### Author Response · Authors · 2023-11-19
> **Response to Reviewer r6Zr (Part 1)**
>
> Thanks for the helpful comments. Below, we address the raised concerns.
>
> ## Connection between Proposition 4.1 and Eq. 1.... It is unclear to me how Proposition 1 and ECE inform the design choices of the method.
>
> We apologize for the confusion. Please allow us to clarify.
>
> In score-based generative modeling, accurate estimation of the score function is of critical importance.
> However, for off-the-shelf classifiers, the second-order score information is difficult to access.
> Proposition 4.1 states that if the calibration error is controlled, so is the fisher divergence.
> Since the calibration error can be estimated by the so-called expected calibration error (ECE), we turn to it as our criterion for tuning the classifier. Equation 1 is the definition of the expected calibration error, which counts for a certain range of predicted confidence, how much does it agree with the actual accuracy.
>
> Taking the diffusion process into consideration, we propose to look at the integral calibration error and it has been one of our biggest motivations for the design choices.
>
>
> ## I think the storyline of this paper is a little bit scattered.
>
> The integral calibration error, denoted as $\overline{\text{ECE}}$, plays a crucial role in our research. It assesses the calibration of the off-the-shelf classifier throughout the diffusion process. Proposition 4.1 establishes a connection between classifier guidance and probability estimation, emphasizing the significance of achieving a lower calibration error. In Figure 1, we analyze $\overline{\text{ECE}}$ values and demonstrate that the off-the-shelf ResNet achieves lower $\text{ECE}_t$ values than the fine-tuned classifier when the noises are large. This finding provides a strong basis for asserting that the off-the-shelf classifier has the potential to match the performance of fine-tuned classifier guidance.
>
> To leverage this potential, we utilize the predicted denoised image $\hat{x}_0(t)$ as the input to the classifier, as suggested in universal guidance [Bansal et al. 2023]. To achieve improved calibration, we propose three key design elements: smooth classifier guidance, consistent guidance direction, and an updated guidance schedule. These design choices aim to calibrate the off-the-shelf classifier with the diffusion models and enable the generation of high-quality guided samples. By incorporating these elements, we enhance the calibration of the off-the-shelf classifier and facilitate the production of superior samples.
>
> We have refined the writing at the end of Section 4.1.
>
> ## "Ablation study is missing"
>
> Thanks for the question. From Section 4.2 to Section 4.5, we have provided ablation studies for each of the design choices.
> We agree that it's important to look at all of them together in one task. Hence, we have provided additional ablation experiments.
>
> The table below (Table 6 of our revised paper) showcases the results of our off-the-shelf classifier guided sampling in sequential ablation experiments, where we explore different design choices. The sequential parameter rule is employed to identify the selection that yields the best FID value. The percentage number within the brackets of each row indicates the improvement achieved by each design choice compared to the baseline setting, which is represented in the first column of each row. The evaluation process involves generating 10,000 samples from the 128x128 ImageNet dataset using ADM[4], utilizing 250 DDPM steps.
>
> Table: Sequential ablation:
> | Inputs |   $\hat{x}_t$   | **$\hat{x}_0(t)$** |  |  |   |
> |---|---|---|---|---|---
> | FID    |  8.61  | **7.17 (16\%)**    |   |   |
> | Softplus $\beta$ | inf (ReLU) |  5.0 | 4.0 | **3.0** |  2.0 |
> | FID              | 7.17 |  6.89  |  6.73  |  **6.61 (8\%)** | 7.02  |
> | marginal temperature $\tau_2$| 1.0 | 0.8 | 0.7 | **0.5** | 0.3 |
> | FID                          | 6.61 |  5.62  | 5.45 | **5.27 (20\%)** | 5.30 |
> | sin factor $\gamma$ | 0.0 | 0.1 | 0.2 | **0.3** | 0.4 |
> | FID                 | 5.27 | 5.30  | 5.27 | **5.24 (1\%)** | 5.38 |
>
> We have added the ablation study in Section 4.6 of our revision.

---

> > ### Author Response · Authors · 2023-11-19
> > **Response to Reviewer r6Zr (Part 2)**
> >
> > ## Question 1: "Which sample did the authors use when providing the ECE results for the rest of the experiments?"
> >
> > To estimate the integral calibration error ($\overline{\text{ECE}}$), we utilize the predicted probabilities from the off-the-shelf classifier obtained through reverse process samples at each timestep of the classifier-guided diffusion sampling. These predicted probability values are used to calculate $\overline{\text{ECE}}$, which serves as a metric to evaluate the calibration performance of both the classifier and diffusion models throughout the guided sampling process.
> >
> > ## Question 2: "It is unclear to me how did the authors incorporate the Softplus activation function into the pre-trained classifiers."
> >
> > We just replace the original ReLU activation functions with Softplus activation in the pre-trained ResNet classifier, eliminating the need for any additional training. This approach has been explored in [Zhu et al. 2021] for improving adversarial transferability.
> >
> >
> > ## Question 3: "How did the authors calculate the marginal likelihood for CLIP guidance generation?"
> >
> > During the CLIP-guided diffusion process, we solely rely on the gradient of the joint likelihood as the guidance signal. This approach is adopted due to the absence of a marginal class in CLIP models, which influences our decision to utilize only the joint likelihood gradient for guidance.
> >
> > It is correct that some of our design choices only apply to probabilistic models with assessable likelihood.
> > Nevertheless, our proposed new guidance schedule (Section 4.5) is proven effective for CLIP-guided generation.
> > Figure 6 of our revised paper demonstrates a comparison of the CLIP scores between the original guidance schedule and our proposed schedule during the sampling process. The results clearly indicate that our updated schedule consistently can achieve significantly higher CLIP scores throughout the entire sampling process.
> >
> > ## Question 4: "What is the recurrent step in Table 1? Why is the inference time not changed significantly with the calibrated method given there is extra marginalization required for all classes? What type of GPU did the authors use in their experiments?"
> >
> > We apologize for the confusion.
> > The recurrent step refers to the backward/forward guidance technique employed in the universal guidance paper [Bansal et al. 2023].
> > Simply put, if the classifier gradient is inferenced 3 times per step of the diffusion denoising process, the recurrent step is 3.
> >
> > Our calibrated designs can all be seen as a pre-conditioning step, which involves replacing the activation functions, changing the softmax temperature, etc. They do not introduce additional computational overhead, thus inference time remains unchanged.
> > The purpose of increasing the number of recurrent steps is to demonstrate that augmenting the guidance steps alone does not substantially improve sampling quality. In contrast, our well-calibrated designs leverage the off-the-shelf classifier to enhance sampling performance without extending sampling time.
> >
> > Regarding "marginalization," it typically refers to the exponential sum of logits of all classes, which is the denominator term used in the softmax function: $\frac{\exp(\tau f_y(x))}{\sum^N_{i=1} \exp(\tau f_i(x))}$. This traditional calculation is commonly employed in classifier-guided sampling [Dhariwal et al. 2021] and is not an additional computation we introduced.
> >
> > For all of our experiments, we use NVIDIA V100 GPUs. We have added this information in the caption of Table 1 in our revision.

---

> > > ### Comment · Reviewer_r6Zr · 2023-11-21
> > > **Response to Rebuttal Part 2**
> > >
> > > 1. **Which sample is used for calculating ECE:** Please allow me to rephrase my question: For reporting ECE values, is the sample $x_t$ or $\hat{x_0}(t)$ used in the classifier?
> > > 2. **How to use Softplus:** Thank you. This answers my question.
> > > 3. **Marginal Likelihood for CLIP:** Thank you. This answers my question, and I would encourage the authors to include this detail in their manuscript if they haven't done that already.
> > > 4. **Details about the experiments:** Thank you and this answers my question.

---

> > ### Comment · Reviewer_r6Zr · 2023-11-21
> > **Response to Rebuttal Part 1**
> >
> > Thank you for your response. While some of my questions are addressed in authors' response, there are still further clarification I would like to request from them:
> >
> > 1. **Connection between Proposition 4.1 and Eq. 1:** Thank you for your clarification. This answers my question.
> > 2. **How does ECE inform the design choices:** Although the authors clarified the relationship between Proposition 4.1 and their motivation, which has been very helpful, unfortunately they didn't answer my second concern directly. Can the authors provide more details about how ECE inform their design choices for increasing the smoothness and marginal temperature? (If you have mentioned it in your modified manuscript, please correct me and point me to the location where you mentioned it)
> > 3. **Ablation Study**: Thank you for the additional experiment. This has addressed my concerns. A minor suggestion is to include the type of evaluation (i.e. FID) in your manuscript because the current text doesn't mention that.

---

> ### Comment · Reviewer_r6Zr · 2023-11-21
> **Overall response to the rebuttal**
>
> Although the authors answered many of my questions, I would still like the authors to clarify how they choose their design choices based on ECE before I raise my score. But overall I am inclining to raise my score and also adjust my confidence to 3 in case there is any further misunderstanding from my side.

---

> > ### Author Response · Authors · 2023-11-22
> > **Response to Reviewer r6Zr**
> >
> > We really appreciate the detailed feedback. Please allow us to provide further clarifications to the unresolved concerns.
> >
> > ## "How does ECE inform the design choices"
> > ### Smoothness
> > The link between ECE and smoothness is twofold, as we have explained in the beginning of Section 4.3. Below, we provide further clarification.
> >
> > First, notice that one of the assumptions for Proposition 4.1 to hold is that the classifier satisfies certain smoothness conditions (inside a Sobolev space).
> > If the activation function is ReLU, the classifier will be piecewise linear with first-order derivatives being piece-wise constant. Not only will it **violate the smoothness assumption** of Proposition 4.1, but it is also bad for **gradient-based optimization**.
> > Therefore, we considered changing the ReLU activation function to smoothed ones such as Softplus.
> >
> > Second, while trying different choices of the parameter $\beta$ within the Softplus function, we found that it also has a **significant effect on ECE**. Hence, we propose to directly choose $\beta$ by minimizing the ECE. We found the $\beta$ that minimizes the ECE also minimizes the generation FID (Please see the results in Figure 3). To be best of the authors' knowledge, tuning activation function for classifier calibration has not been explored before.
> >
> > To summarize, **Proposition 4.1 links classifier smoothness, calibration, and gradient estimation together.**
> > By modifying the activation function of off-the-shelf classifiers, we are able to achieve both improved smoothness and calibration.
> >
> >
> > ### Marginal temperature
> > As for the marginal temperature, minimizing ECE is not its motivation. As stated in Section 4.4, this design choice is to **balance the joint vs. conditional direction of the classifier's gradient**.
> >
> > On a high level, the joint direction points toward the distribution mode while the conditional direction points away from the decision boundary.
> > Inspired by Proposition 4.2 and the visualization in Figure 4, we **amplify the joint direction** relative to the marginal one to help guide the overall diffusion generation process toward the mode of
> > density.
> >
> > This is further supported by the ablation study presented in Table 4 of our paper, where reducing values of the marginal temperature ($\tau_2<1$) exhibit **significant improvements** over the baseline ($\tau_2=1$) in terms of FID.
> > By choosing $\tau_2=0.5$ improves the performance (FID) by **20\%** over the baseline.
> >
> > We provide further visualization in Figure 4 of our paper. The left-hand side figure illustrates the traditional guidance setting, where the classifier gradient figure gradually fades from $t=50$ to 0, indicating a loss of dog depiction guidance during the sampling. In contrast, in the right-hand side figure where the joint guidance was amplified, the classifier gradient figure increasingly highlights the dog's outline, providing consistent and accurate guidance direction throughout the entire sampling process.
> >
> > ## "Which sample is used for calculating ECE?"
> > In our pilot experiment, Figure 1 compares the ECE of the fine-tuned and off-the-shelf classifiers calculated based on $\hat{x}_t$.
> >
> > When considering classifier-guided diffusion generation, both $\hat{x}_t$ and $\hat{x}_0(t)$ are viable choices.
> > In Table 2 of our revised paper, both choices are tested for integral ECE estimation.
> > The results suggest that it is better to use $\hat{x}_0(t)$ as input to the classifier due to its superior guidance performance. Thus, **we stick to the $\hat{x}_0(t)$ as classifier inputs** for integral ECE estimation in the following analysis such as Table 3.
> >
> > We hope that the above clarifications could resolve your remaining concerns.

---

> ### Comment · Reviewer_r6Zr · 2023-11-22
> **Thank you for your response**
>
> Thank you for your response. These answers clarify all my questions, and although I still think the story of the paper can be better structured since not all the tricks/methods developed are related to the main claimed contribution of the paper (i.e. using calibration as the general guideline to analyze the design space of classifier guidance diffusion), the analysis presented in this paper is significant enough to warrant an acceptance. Therefore, I am increasing my score to a 6.

---

### Official Review · Reviewer_6jri · 2023-11-08

**Soundness:** 4 excellent
**Presentation:** 3 good
**Contribution:** 3 good
**Rating:** 8
**Confidence:** 5

**Summary:**

The paper investigates improving sample quality in conditional diffusion generation by leveraging off-the-shelf classifiers in a training-free context. The authors propose pre-conditioning techniques based on classifier calibration, significantly enhancing diffusion models' performance with minimal computational overhead, verified through experiments on ImageNet. They introduce a novel metric, integral calibration error (ECE), to evaluate the effectiveness of classifier guidance and find that off-the-shelf classifiers outperform trained classifiers under high noise. To combat the diminishing influence of classifier guidance in later diffusion stages, a new weighing strategy is suggested, yielding better sample quality. The paper highlights the potential of their methods in text-to-image generation and points out the limitations of current guidance enhancement methods in terms of sampling efficiency.

**Strengths:**

- The paper convincingly establishes the research context. In particular, it is effective in demonstrating the robustness differences across time step intervals between off-the-shelf classifiers and fine-tuned classifiers using the ECE metric.
- The division of the design space for diffusion guidance appears appropriate, and the empirical process of selecting among the various options seems justified.
- The authors' exploration reveals that guidance using off-the-shelf models exhibits performance comparable to or surpassing previous methods, which required significant computational costs.
- The experiments with guidance through CLIP demonstrate the potential for extending guidance models beyond classifiers, indicating scalability in the approach.
- The theoretical explanations provided lend solid persuasive strength to the authors' design choices.

**Weaknesses:**

- While I rate this study highly in general, it falls short in comparing with previous research utilizing off-the-shelf models.
- Specifically, the omission of closely related prior works, [1, 2] , is significant. PPAP[2] explores guidance using a wide range of modalities of off-the-shelf models in an efficient tuning and plug-and-play manner without significant computational costs, which is directly relevant to this paper's discussion. Observations such as the varying contributions of tuning guidance models across different time steps could also reinforce the evidence between the two studies.
- Overall, I highly appreciate the paper's contribution to the completeness of the discussion on diffusion models' guidance. However, to properly highlight this paper's direct contributions, a comparison with prior studies attempting off-the-shelf guidance is crucial. Although the authors conducted experiments comparing various design choices of off-the-shelf guidance, it is not clearly presented how these relate to previous research, and there is no direct comparison of the final FID scores with prior methodologies. The lack of such comparisons has inclined my evaluation towards rejection.
- In the CLIP guidance experiments, it seems that the authors did not apply the same level of guidance as they did with off-the-shelf classifiers. The CLIP guidance experiment does not appear to reflect the authors' contributions with a new methodology for off-the-shelf guidance.

[1]: Graikos, Alexandros, et al. "Diffusion models as plug-and-play priors." Advances in Neural Information Processing Systems 35 (2022): 14715-14728.

[2]: Go, Hyojun, et al. "Towards practical plug-and-play diffusion models." Proceedings of the IEEE/CVF Conference on Computer Vision and Pattern Recognition. 2023.

**Questions:**

- Based on the various considerations and discoveries made by the authors, I am curious whether it is possible to extend off-the-shelf guidance beyond class conditions to various modalities of conditional generation. It seemed that the CLIP experiment was intended to demonstrate such a possibility; however, as mentioned in the weaknesses, it did not feel like an experiment based on a new methodology reflecting the authors' considerations and findings.

---

> ### Author Response · Authors · 2023-11-19
> **Response to Reviewer 6jri - Part 1**
>
> Thanks for the helpful comments. Below, we address the raised concerns.
>
> ## Related work: the omission of closely related prior works, [1, 2], is significant...  Although the authors conducted experiments comparing various design choices of off-the-shelf guidance, it is not clearly presented how these relate to previous research, and there is no direct comparison of the final FID scores with prior methodologies.
>
>
> Thanks for pointing out the related works [1] and [2]. Both works investigated guided diffusion generation. However, their focus is different from ours, and hence their proposed methodologies are fundamentally different. Below we provide a detailed comparison to each of them.
>
>
> The goal of [1] is to demonstrate the flexibility of conditional diffusion generation given specified constraints by introducing a regularization term $\log c(x, y)$ in the denoising process. Specifically, it explores constraints to control the digit shape in MNIST generation, facial specified attributes, infer semantic segmentations of city images, and solve a continuous relaxation of the traveling salesman problem.
>
> The take-home message of this work is that diffusion models, trained on domain-specific data and under carefully designed conditions, are **capable of generating samples that conform to specified constraints**. While this showcases the flexibility of diffusion models in producing constraint-compliant samples, it **doesn't test the limit of the performance in each of the categories**.
>
> In comparison, we specifically investigate class-conditional image generation and demonstrated that with proper designs, off-the-shelf classifiers can beat existing state-of-the-art methods.
> Through comprehensive experiments on different architectures of diffusion models, including DDPM, EDM, and DiT, our off-the-shelf classifier-guided sampling consistently surpasses the baseline performance across various diffusion model architectures.
>
> Specifically, in the table below (Table 7 of our revised paper), our FID result of 2.19 on ImageNet128x128 not only surpasses the fine-tuned classifier-guided performance (FID: 2.97) [4], but also outperforms classifier-free trained diffusion models (FID: 2.43) [5]. This highlights that off-the-shelf guidance is not merely a substitute for fine-tuned guided diffusion to save training costs, but rather a better choice for generating high-quality samples.
>
> Table: ImageNet 128x128 generation results.
> |                 |   IS   |   FID  |  sFID | Precision| Recall |
> |---|---|---|---|---|---|
> | ADM             |    -   |  5.91  | 5.09  | 0.70  | 0.65
> | ADM-G           | 182.69 |  2.98  | 5.09  | 0.78  | 0.59
> | CFG             | 158.47 |  2.43  | -     |   -   |  -
> | Resnet101-guided (Ours)| 187.83 |  2.19  | 4.53  | 0.79  | 0.58
>
>
> [2] introduces a multi-expert strategy that utilizes multiple expert classifiers, each trained to handle a specific range of noise values. These "experts" guide the diffusion sampling process at their corresponding time steps. The authors propose a parameter-efficient knowledge transfer strategy to reduce the computational cost. Further, they proposed a teacher-student training approach to avoid the need for labeled data.
>
> However, the experiments conducted in [2] do not provide conclusive evidence that multi-expert-guided sampling can compete with the fine-tuned classifier-guided diffusion ADM-G [4].
> Moreover, though they employed parameter-efficient fine-tuning strategies like LORA, **their computation cost is still non-negligible and cannot be used in a plug-and-play fashion**.
>
> In contrast, our off-the-shelf guided sampling not only eliminates the need for further training but also delivers significantly higher sampling performance. As demonstrated in Table 7 of our revised paper, our FID result of 2.19 on ImageNet128x128 not only surpasses the performance of the fine-tuned classifier-guided approach (FID: 2.97) [4], but also outperforms diffusion models trained without the use of classifiers (FID: 2.43) [5].

---

> > ### Author Response · Authors · 2023-11-19
> > **Response to Reviewer 6jri - Part 2**
> >
> > ## CLIP score："In the CLIP guidance experiments, it seems that the authors did not apply the same level of guidance as they did with off-the-shelf classifiers. The CLIP guidance experiment does not appear to reflect the authors' contributions with a new methodology for off-the-shelf guidance."
> >
> > In the CLIP-guided diffusion section, we showcase the possibilities for future research in exploring other modalities. Figure 6 of our revised paper provides a comparison of the CLIP scores between the original guidance schedule and our proposed schedule during the sampling process. The results clearly indicate that our updated schedule consistently can achieve significantly higher CLIP scores throughout the entire sampling process.
> >
> >
> > ##  "I am curious whether it is possible to extend off-the-shelf guidance beyond class conditions to various modalities of conditional generation. "
> >
> > This research primarily focuses on exploring classifier guidance, specifically in the context of text-to-image generation using an off-the-shelf CLIP model. However, in our future research endeavors, we intend to extend our investigation to encompass guidance from other modalities, such as leveraging the capabilities of large language models. By exploring these additional modalities, we aim to further enhance the performance and flexibility of our proposed approach.
> >
> > We believe this is a very important future direction to explore. Currently, the major concern for state-of-the-art text-to-image models such as Stable Diffusion is not image quality, but controllability, i.e., the compatibility between the generated image and the input text.
> > For instance, a simple prompt like "a green flower and a yellow leaf" is very challenging [7].
> >
> > T2I-compbench is specifically designed to quantitatively measure text-image compatibility, which employs several image-to-text discriminative models, e.g., BLIP-VQA, to serve as the judge.
> > This indicates that SOTA image-to-text discriminative models are better in terms of text-image compatibility, and have the potential to be used as guidance to improve the text-to-image generation.
> > Our off-the-shelf guidance method can potentially be very helpful in improving the controllability of text-to-image generation.
> >
> > For all modalities, evaluating the generated samples is a challenging task. We have FIDs for images but such a well-accepted criterion is missing for 3D generation.
> > Nevertheless, as long as there are powerful discriminative models that can tell whether the generated samples are good or not, they can be utilized in a training-free fashion by extending our method to achieve a significant boost in conditional generation across every modality.
> >
> >
> > Reference:
> >
> > [1]: Graikos, Alexandros, et al. "Diffusion models as plug-and-play priors." Advances in Neural Information Processing Systems 35 (2022): 14715-14728.
> >
> > [2]: Go, Hyojun, et al. "Towards practical plug-and-play diffusion models." Proceedings of the IEEE/CVF Conference on Computer Vision and Pattern Recognition. 2023.
> >
> > [3]: Bansal, Arpit, et al. "Universal guidance for diffusion models." Proceedings of the IEEE/CVF Conference on Computer Vision and Pattern Recognition. 2023.
> >
> > [4]: Dhariwal, Prafulla, and Alexander Nichol. "Diffusion models beat gans on image synthesis." Advances in neural information processing systems 34 (2021): 8780-8794.
> >
> > [5]: Ho, Jonathan, and Tim Salimans. "Classifier-free diffusion guidance." arXiv preprint arXiv:2207.12598 (2022).
> >
> > [6]: Nichol, Alex, et al. "Glide: Towards photorealistic image generation and editing with text-guided diffusion models." arXiv preprint arXiv:2112.10741 (2021).
> >
> > [7] Huang, Kaiyi, et al. "T2i-compbench: A comprehensive benchmark for open-world compositional text-to-image generation." NeurIPS 2023.

---

> > > ### Comment · Reviewer_6jri · 2023-11-21
> > >
> > > The authors have satisfactorily addressed all the concerns I had raised. I trust that the discussions, particularly regarding the previously omitted comparisons and citations of related work, will be diligently incorporated into the final revision. I extend my thanks to the authors and am inclined to adjust my final rating towards acceptance.

---

### Author Response · Authors · 2023-11-19
**Revision Summary**

We thank all the reviewers for their helpful comments. We have addressed them individually in our rebuttal response.
Additionally, we have made the following revisions in our submitted PDF file:

* In Section 2, we added more in-depth discussions of related works.

* In Section 4.1, we added more discussion to further clarify the connection between classification calibration and classifier-guided diffusion generation.

* In Section 4.5, we fixed the notation of the $\gamma_t$ schedule and added more intuition behind this design.

* We added a new Section 4.6 to include the sequential ablation study for all of our proposed design choices. We have included the percentage of improvement achieved by each design choice compared to the baseline setting.

* In Section 5.4, we added the CLIP-guided generation experiment (Figure 6), it demonstrates significantly higher CLIP scores under our proposed schedule than the baseline throughout the generation process.

---

### Author Response · Authors · 2023-11-23
**Rebuttal Summary**

We really appreciate the time and efforts from all the reviewers and the AC.

In this work, we showcased **for the first time** that with proper designs, it is possible to achieve better performance in guided diffusion generation using off-the-shelf classifiers than the typical CG or CFG methods, despite they are trained/finetuned with the diffusion model.
Our experiments are mainly on the class-conditional generation setting, which we think is **more convincing** since the quantitative benchmarks are **more competitive and saturated**.
Achieving such surprising results requires in-depth investigations into the synergy between the classifier and the diffusion process, which is **non-trivial**.

To demonstrate generalizability, we also showcased a significant boost in CLIP-guided diffusion generation (Figure 6 in our revision). With the proliferation of pre-trained discriminative models in various domains and modalities, our investigation has implications that the plug-and-play, training-free guidance method can be **state-of-the-art in conditional generation across every modality**.



During the rebuttal period, we have provided further clarifications to our approach, including discussions of related works, motivations for each of the design choices, implementation details, ablation results, etc.
The reviewers' questions and constructive comments provide valuable guidance for us to improve the writing and make a more convincing case in our revision.

---

### Meta-Review · Area_Chair_b2N3 · 2023-12-14

**Metareview:**

This work explores the potential of enhancing diffusion generation in image synthesis through the use of off-the-shelf classifiers. The authors investigate the application of pre-conditioning techniques and calibration to improve sample quality and control in diffusion models without additional training. The reviewers in general commended the novel approach of using off-the-shelf classifiers for guided diffusion. They appreciated the empirical evidence supporting the superiority of this method over existing classifier guidance and CFG methods. Some reviewers raised concerns about the direct comparison with related prior works and the generality of the approach across various modalities. Questions were raised about the methodology's connection to calibration error and how it informed the design choices. The authors addressed most of the concerns, for example, they clarified how expected calibration error (ECE) influenced their design choices, particularly in smooth classifier guidance and joint vs. conditional direction. They also provided additional ablations and explanations of their approach's effectiveness in various settings, including with the CLIP model. Despite borderline/negative ratings initially, after the rebuttal, several reviewers increased their ratings to a positive score of 6.

Overall, the AC concurs with the reviewers that this study has the potential to provide useful knowledge advancement in the field of image generation. AC also agrees that the possibility of extending this work to text-to-image synthesis represents an exciting direction for future research and does not hinder the current submission's suitability for acceptance.

**Justification For Why Not Higher Score:**

Overall, the reviewers exhibit a neutral to positive stance towards the proposed method, yet there is a noticeable absence of strong enthusiasm. This tempered response is primarily attributed to several crucial areas that constrain the possibility of a higher score. These include the need for a more robust comparison with previous studies, enhanced clarity and depth in the methodology, and the incorporation of more comprehensive evaluation metrics.

**Justification For Why Not Lower Score:**

Again, the reviewers' response to the paper is neutral to positive. They haven't identified any significant flaws or drawbacks, suggesting that accepting the paper could be beneficial to the wider community engaged in research on guided diffusion models.

---

### Decision · Program_Chairs · 2024-01-16

Accept (poster)